

# Impact of model improvements on 80-m wind speeds during the second Wind Forecast Improvement Project (WFIP2)

Laura Bianco[1,2], Irina V. Djalalova[1,2], James M. Wilczak[2], Joseph B. Olson[1,2], Jaymes S. Kenyon[1,2], Aditya Choukulkar[1,2], Larry K. Berg[3], Harindra J. S. Fernando[4], Eric P. Grimit[5], Raghavendra Krishnamurthy[4], Julie K. Lundquist[6,7], Paytsar Muradyan[8], Mikhail Pekour[3], Yelena Pichugina[1,2], Mark T. Stoelinga[5], David D. Turner[2]

[1]University of Colorado/Cooperative Institute for Research in Environmental Sciences, Boulder, CO, USA
[2]National Oceanic and Atmospheric Administration/Earth Systems Research Laboratory, Boulder, CO, USA
[3]Pacific Northwest National Laboratory, Richland, WA, USA
[4]Civil and Environmental Engineering and Earth Sciences, University of Notre Dame, Notre Dame, IN, USA
[5]Vaisala Inc., Seattle, WA, USA
[6]Department of Atmospheric and Oceanic Sciences, University of Colorado Boulder, Boulder, CO, USA
[7]National Renewable Energy Laboratory, Golden, CO, USA
[8]Argonne National Laboratory, Lemont, IL, USA

*Correspondence to*: Laura Bianco (laura.bianco@noaa.gov)

**Abstract.** During the second Wind Forecast Improvement Project (WFIP2; Oct 2015 – Mar 2017, Columbia River Gorge and Basin area) several improvements to the parameterizations applied in the High Resolution Rapid Refresh (HRRR – 3 km horizontal grid spacing) and the High Resolution Rapid Refresh Nest (HRRRNEST – 750 m horizontal grid spacing) Numerical Weather Prediction (NWP) models were tested during four 6-week reforecast periods (one for each season). For these tests the models were run in control (CNT) and experimental (EXP) configurations, with the EXP configuration including all the improved parameterizations. The impacts of the experimental parameterizations on the forecast of 80-m wind speeds (hub height) from the HRRR and HRRRNEST models are assessed, using observations collected by 19 sodars and 3 profiling lidars for verification. Improvements due to the experimental physics (EXP vs CNT runs) versus those due to finer horizontal grid spacing (HRRRNEST vs HRRR), and the combination of the two are compared, using standard bulk statistics such as Mean Absolute Error (MAE) and Mean Bias Error (bias). On average, the HRRR 80-m wind speed MAE is reduced by 3-4% due to the experimental physics. The impact of the finer horizontal grid spacing in the CNT runs also shows a positive improvement of 5% on MAE, which is particularly large at nighttime and during the morning transition. Lastly, the combined impact of the experimental physics and finer horizontal grid spacing produces larger improvements in the 80-m wind speed MAE, up to 7-8%. The improvements are evaluated as a function of the model's initialization time, forecast horizon, time of the day, season of the year, site elevation, and meteorological phenomena, also looking for the causes of model weaknesses. Finally, bias correction methods are applied to the 80-m wind speed model outputs to measure their impact on the improvements due to the removal of the systematic component of the errors.



# 1 Introduction

The second Wind Forecast Improvement Project (WFIP2) took place in Oregon and Washington states from October 2015 through March 2018. This Department of Energy (DOE) and National Oceanic and Atmospheric Administration (NOAA) funded project was aimed at improving the parameterizations within the High Resolution Rapid Refresh (HRRR – 3 km horizontal grid spacing) and its nested version (HRRRNEST – 750 m horizontal grid spacing), with the goal of increasing the forecast skill of hub-height wind speeds. The study area is a region of complex terrain that included a large amount of wind power generation, with more than 4.6 GW of installed capacity associated with the Bonneville Power Administration (BPA) balancing authority.

WFIP2 (Olson et al., 2019a; Shaw et al., 2019; Wilczak et al., 2019a), as well as the first WFIP (held in the U.S. Great Plains, in 2011-2012; Wilczak et al., 2015), represent efforts to improve forecasts for the renewable energy sector. While the first WFIP was in an area with relatively flat terrain, WFIP2 took place in an area characterized by pronounced topographic features. These include the Cascade Mountains and the Columbia River Basin to the east, with the Columbia River Gorge forming a gap in the mountain range owing to complex flow patterns in the region. Important background information regarding the project can be found in several publications: Shaw et al. (2019) presents a general overview of the project; Wilczak et al. (2019a) describes the instruments deployed for the 18-month long campaign and the meteorological forecast challenges of the region; and Olson et al. (2019a) discusses the parameterization improvements applied to the HRRR and HRRRNEST models resulting from a better understanding of local atmospheric processes achieved by the use of the observations.

Toward the end of the campaign, a model freeze was imposed and four 6-week periods (one for each season: "spring 2016" – 3/25-5/7/2016, "summer 2016" – 6/24-8/7/2016, "fall 2016" – 9/24-11/7/2016, and "winter 2017" – 12/25/2016-2/7/2017) were chosen to re-run the models in control (CNT) and experimental (EXP) configurations. The EXP configuration included all the modifications/improvements added to the models, while the CNT runs used the HRRR parameterization present in the NCEP operational version of the HRRR at the start of WFIP2. The four 6-week periods will be called "reforecast periods" throughout the rest of the manuscript, while the model re-runs (HRRR CNT, HRRR EXP, HRRRNEST CNT, and HRRRNEST EXP) will be called "reforecast runs".

Since the primary goal of WFIP2 is to advance the state of the art of wind energy forecasting in areas with complex terrain in general, and in the BPA region in particular, in this paper we use hub-height wind speed observations from sodars and profiling lidars to assess the impacts of the experimental parameterizations and finer horizontal grid spacing on the performance of the models. These instruments were chosen because they accurately measure wind speed and direction from 20 m up to few hundred meters above ground level, which is the layer of the atmosphere most relevant for wind energy production. While in this paper improvements in bulk statistics (Mean Absolute Error – MAE, and bias) are evaluated, a companion research article (Djalalova et al., 2019) determines the improvements using the same set of measurements and the same model runs at forecasting wind power ramp events.





The paper is organized as follows: in Section 2 the observation and NWP model datasets are described; in Section 3 details of the bulk statistical results are presented for 80-m wind speed MAE and bias for individual models, in terms of time of the day, model initialization time, forecast horizon, season of the year, and site elevation; in Section 4 improvements in the statistical results are quantified due to the experimental physics, model finer horizontal grid spacing, and a combination of the two, again

as a function of the time of the day, the season of the year, and the different meteorological phenomena predominant in the area, both with and without bias correcting the model output. Section 5 presents summary and conclusions.

## 2 Dataset description

### 2.1 Observational dataset

Various in-situ, scanning, and profiling instruments were deployed and maintained by WFIP2 team partners who later provided

quality controlled versions of the data. All data are available to the public from the DOE Data Archive and Portal (DAP; https://a2e.energy.gov/projects/wfip2). The list of instruments, deployed in nested arrays (with the outer scale of the order of 500km and the inner scale of the order of 2x2km, see Fig. 1a of Wilczak et al., 2019a), includes 3 449-MHz, 8 915-MHz radar wind profilers with radio acoustic sounding system temperature profiles, 19 sodars, 5 scanning lidars, 5 profiling lidars, 4 microwave radiometers, 10 microbarographs, a network of sonic anemometers, and many surface meteorological stations.,

includes 3 449-MHz, 8 915-MHz radar wind profilers with radio acoustic sounding system temperature profiles, 19 sodars, 5 scanning lidars, 5 profiling lidars, 4 microwave radiometers, 10 microbarographs, a network of sonic anemometers, and many surface meteorological stations. An overview of the instrumentation capability and how the instruments were used for atmospheric process understanding and model verification and validation is presented in Wilczak et al. (2019a) and Olson et al. (2019a). In the current study, data collected at 22 remote sensing sites (19 sodars and 3 lidars) spanning the WFIP2 region

are used, since their measurements cover the part of the atmosphere of most interest for wind energy. As measurements through the entire turbine-rotor layer were not always available, we decided to focus on the 80-m level when available, to avoid averaging the data over a variable depth layer of the atmosphere that could result, in some cases, to biasing the average toward values more representative of the lower part of the layer.

Some sites had a co-located sodar and lidar. In this situation the instrument with the highest data availability during the

campaign was chosen. This choice led to the selection of the 19 sodars and 3 lidars listed in Table 1, where the latitude, longitude, elevation of the site, terrain complexity, percentage of data availability over the four reforecast periods, and the institution in charge of the instrument are also presented. The terrain complexity was computed as the standard deviation (in meters) relative to the average slope in a 6 by 6 km area (81 points) around the site using the HRRRNEST model topography. Although the focus of this study is on the 80-m wind speed statistics, we also examine the statistics of wind power generation,

using a generic IEC Class 2 power curve to convert wind speed into power. Details for the conversion from wind speed into power are given in Wilczak et al. (2019b), while Wilczak et al. (2019a) and Djalalova et al. (2019) demonstrated that the equivalent wind power generation computed from these 22 remote sensors using the above mentioned curve is representative



of the actual wind power generation over the entire BPA area. The geographical location of the 19 sodars and 3 lidars is provided in Fig.1 of Djalalova et al. (2019), and a more comprehensive base map of all the instruments deployed for WFIP2 is presented in Wilczak et al. (2019a).

## 2.2 NWP Models

The HRRR and the HRRRNEST are the models of interest in this study due to their small horizontal grid spacing, which is better suited for an area of complex terrain such as WFIP2. For the four reforecast periods (spring, summer, and fall 2016, and winter 2017), 24-hour forecasts were made with the HRRR and HRRRNEST, with output every 15 minutes, using initial conditions from the operational RAPid refresh model (RAP; Benjamin et al., 2016), with no additional data assimilation. The models were run twice a day, at 0000 UTC and 1200 UTC. For simplicity, we refer to the runs initialized at 0000 UTC as the

Z00 runs, and at the runs initialized at 1200 UTC as the Z12 runs. The HRRRNEST output runs were delayed by 3 hours to avoid spin-up problems, so that a gap in the HRRRNEST model output exists from forecast horizon 0000 to forecast horizon 0200 (from 0000 UTC – 0200 UTC for the Z00 initialized runs, and from 1200 UTC – 1400 UTC for the Z12 initialized runs). For this reason, in order to show meaningful comparisons between the models, we utilize only the forecast horizons 03-24 for the HRRR runs.

The reforecasts were run in both CTL and EXP configurations. Differences between the two include added parameterizations to the HRRR and HRRRNEST physics suite (i.e. representation of wind farms and of drag associated with subgrid-scale (SGS) topography in the HRRR), improvements to existing parameterizations (i.e. boundary-layer and surface-layer schemes, cloud–radiation interaction), and improvements to numerical methods (i.e. finite differencing of the horizontal diffusion). The biggest improvements were from the boundary-layer, finite differencing, and the SGS drag, which improved the representation of

turbulent mixing in stable boundary layers. The reader is referred to Olson et al. (2019a; 2019b) for more details on the differences between the CTL and EXP model configurations. For our analysis, in order to compare to the observations, the 80-m wind field model output is horizontally bi-linearly interpolated to the 22 site locations using the 4 closest grid points. The HRRR has relatively coarse vertical resolution, with only five full model layers below 200 m, but the middle of the third layer is very close to 80-m AGL, so a linear interpolation does not negatively impact the accuracy in estimating 80-m wind speeds.

The observations were also averaged and interpolated in time over the 15-minute model output times (most of the observations were already at a 15 min interval, but some were at a 10 min interval or less), and linearly interpolated to the 80-m level.

## 3 Bulk statistical results of 80-m wind speed forecasts

In this section we examine the diurnal variation of 80-m wind speed MAE and bias at all sites and the seasonal variation of MAE and biases from the four reforecast periods to identify the dependence of the statistics on the time of the day, model

initialization time, forecast horizon, and season. The dependence on the elevation of the site is also investigated.



### 3.1 Statistical results as a function of the time of the day, model initialization time, forecast horizon, and season of the year

The 80-m wind speed MAEs, averaged over the 19 sodars and 3 lidars, show a clear diurnal pattern (Fig. 1). Each of the four reforecast runs (HRRR CNT is in red, HRRR EXP in blue, HRRRNEST CNT in yellow, and HRRRNEST EXP in black) is averaged over the four reforecast periods in the upper panel (a), while the lower panels (b-e) show the four reforecast periods separately. Initialization times are represented with the O's (Z00 runs) and with the X's (Z12 runs), while the solid bold lines are the averages between the Z00 and Z12 values. The 80-m wind speed MAEs show a clear diurnal pattern, consistent among all model runs, with larger average MAEs at nighttime (LST = UTC-8) falling mostly between 2 and 2.4 m s$^{-1}$, with smaller values during daytime, ranging between 1.6 and 1.8 m s$^{-1}$ (panel a). In addition to the larger values of MAE found at nighttime, the reforecast runs also show larger differences between the models. In contrast, during daytime not only are the MAEs smaller, but the differences between the four models reforecast runs are also smaller. Examining the dependence of MAE on initialization time and forecast horizon, the Z00 MAEs are smaller than the Z12 MAE values for UTC times soon after the Z00 initialization time (O lines are below X lines), and the Z12 MAEs tend to be smaller than Z00 values for UTC times soon after the Z12 initialization time (X lines are below O lines, except for HRRRNEST EXP), meaning that the MAE increases with the forecast horizon. Between forecast horizon and initialization time it is difficult to separate what has more importance in terms of contributing to MAE values. Certainly, for each of the model reforecast runs, the time of the day is more important at determining the MAE values than either the initialization time or the forecast horizon.

While on average the experimental physics and finer grid spacing lowers the MAEs over the four reforecast periods (Fig. 1, panel a: blue, yellow and black lines all show smaller MAEs compared to the red lines), the improvements are less consistent when looking at the four reforecast periods separately (panels b-e). In winter, the improvements are more robust, as explained in Olson et al. (2019a), due to better maintenance of cold pools which frequently happen in this area over the winter (McCaffrey et al., 2019; Whiteman et al., 2001), and which are investigated in detail in Section 4.4.

The biases of the 80-m wind speed also exhibit a diurnal cycle (Fig. 2). Again, the upper panel shows averages of the four reforecast periods and the lower panels display the four reforecast periods separately. The diurnal trend of the bias in the HRRR CNT is evident in the red curves, with positive biases at nighttime, averaging 0.7 m s$^{-1}$, and negative values during daytime, down to -0.4 m s$^{-1}$ (panel a). The diurnal trend for the HRRR CNT is also clear for the four reforecast periods separately (panels b-e). The HRRR EXP reforecast runs (blue curves) tend to eliminate the diurnal trend in all reforecast periods, but lowers the bias significantly, leading to a negative average value of ~-0.7 m s$^{-1}$ (panel a). A possible reason for such behaviour in the HRRR EXP runs can be found in the representation of drag due to SGS orography (Steeneveld et al., 2008; Tsiringakis et al., 2017) added to the HRRR physics suite. This new representation is only active in the HRRR, but not in the HRRRNEST due to its finer grid spacing (Olson et al., 2019a). While the expected benefit of such improved representation of the drag is to decrease the high wind speed bias in stable conditions often found in the HRRR, the detriment in this case seems to be a too large decrease in wind speed. The addition of wind turbine drag from the wind farm parameterization also contributed to the low wind speed bias, but to a lesser degree. Due to the results found in this study and in other WFIP2 related studies, ways to



revisit the treatment of the drag due to sub-grid scale orography are under consideration. The diurnal trend in the results is much smaller in the winter than in other seasons. This result could be due to differences in the treatment of boundary-layer turbulence in unstable and stable conditions. Similar results were found by Berg et al. (2019) in their study of the sensitivity of winds simulated using the Mellor–Yamada–Nakanishi–Niino planetary boundary-layer parameterization in the Weather Research and Forecasting model.

While the HRRRNEST reforecast runs (CNT in yellow and EXP in black) reduce the bias compared to both HRRR it is not clear yet if the HRRRNEST EXP is better than the HRRRNEST CTL or vice-versa. Similar to the MAEs, differences between the four reforecast runs are larger at nighttime and smaller at daytime (when the biases are consistently mostly negative).

Figure 3 displays the 80-m wind speed MAEs (on the left) and biases (on the right) averaged over all sites, and over all reforecast horizons from 03 to 24, for the four separate reforecast periods. For the MAE of the 80-m wind speed, the HRRR EXP (in blue) does better than the HRRR CNT (in red) in fall and in winter, but not in spring nor summer. MAEs of the HRRRNEST CNT (in yellow) are better than those of the HRRR CNT (in red), and the HRRRNEST EXP (in black) is now almost always better than the other models. Biases in the HRRR EXP (in blue) have become more negative compared to the HRRR CNT (in red), but by too much in the spring, summer and fall. The HRRRNEST EXP (black) is better than the HRRRNEST CNT (in yellow) only in the fall and winter, and again it is not clear that one of these two models has a demonstrably better overall bias.

The results of this section indicate that the time of the day is of primary importance in terms of MAEs and biases, while the model initialization time and the forecast horizon are of secondary importance. Consequently, the remaining statistical analysis is carried out averaging the Z00 and Z12 runs.

## 3.2 Statistical results as a function of the site elevation

As evident from Table 1, the 22 sites used for this analysis have very different elevations (ranging from 63 m asl at Rufus – RFS, to 991 m asl at Prineville - PVE), as well as different surrounding topographic variability. In this section, we investigate the dependence of the model error statistics on the site elevation. In Fig. 4 the results for the 80-m wind speed normalized bias, averaged over the two model initialization times, and over all forecast horizons from 03 to 24, are presented for the four reforecast periods. Sites are sorted from low to high elevation (from Rufus on the left to Prineville on the right) and biases are normalized by the averaged (observed) 80-m wind speed at each site. The biases presented in Fig. 4 show that the diurnally and seasonally averaged biases are smaller (and often negative) at lower elevations, with a positive trend with increasing elevation. In particular, the HRRR CNT (red) has the largest positive bias at high elevations in winter due to the premature mix-out of cold pools, consistent with that described in Wilczak et al. (2019a) and Olson et al. (2019a). As in Fig. 2, HRRR EXP runs (in blue) always show the lowest bias, almost always negative, particularly at the lowest elevation sites. When not normalized by the averaged wind speed at the site (not shown) the trend was consistent with that shown in Fig.4, but even more accentuated. Although it is not clear at this point what is the physical reason for the models having a normalized bias dependent on site elevation (it may be due to the characteristics of the atmospheric phenomena predominant in this area, and

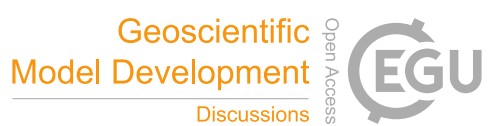

challenging to forecast), it is important to know that in an area of complex terrain like that of WFIP2 this dependence exists. The dependence of the bias on the elevation indicates that a post-processing bias correction of the model should be done at each site independently.

In contrast, a similar analysis but for MAE normalized by the averaged 80-m wind speed at each site (not shown) did show a
5 mostly neutral dependence on site elevation (with a slight decrease with site elevation).

Terrain complexity is not as powerful of a predictor of model bias as site elevation. A similar analysis to that presented in Fig. 4 was performed but sorting the sites by the complexity of the surrounding terrain (see Table 1). In this analysis (not shown) the trend of 80-m wind speed MAE and bias was not clearly defined.

## 4 Improvements to the statistics due to the experimental physics and finer horizontal grid spacing

In this section we examine the statistical significance and percentage improvement in the model forecast of 80-m wind speed and power. The improvements are analyzed in terms of the new physics (EXP vs CNT runs) as well as horizontal grid spacing of the models (HRRRNEST vs HRRR runs), first separately and then combining the impact of the two (HRRRNEST EXP vs HRRR CNT). Finally, we evaluate the dependence of the improvements on the dominant meteorological phenomena of the area (Shaw et al., 2019), including cold pools (McCaffrey et al., 2019; Whiteman et al., 2001; Zhong et al., 2001), gap flows
(Sharp and Mass 2002; 2004), easterly flows (Neiman et al., 2018), mountain waves (Durran 1990; 2003), topographic wakes, and convective outflows (Mueller and Carbone, 1987).

### 4.1 Impact of experimental physics (CNT vs EXP runs)

The impact of the experimental physics in the HRRR runs (HRRR EXP vs HRRR CNT) is almost always positive. Percent improvement and statistical significance is shown in Fig. 5 for 80-m wind speed (left panels) and 80-m wind power (right
panels). These results are obtained averaging all sites together, over the two model initialization times (forecast horizon from 03 to 24), and over the four reforecast periods. The upper panels display the variation of diurnal MAE (HRRR CNT in red and HRRR EXP in blue), and the middle panels show differences between MAEs of the HRRR CNT run and MAEs of the HRRR EXP run. Error bars represent the $\pm 1.96\sigma/\sqrt{n}$ interval of this difference (where the number of points, n, is reduced by the autocorrelation of the models runs), with a 95% confidence level chosen. Finally, the lower panels show the percentage MAE
relative improvement of the HRRR EXP model over the HRRR CNT model (defined as $100 \times$ (MAE HRRR CNT − MAE HRRR EXP) / MAE HRRR CNT). From the bottom panels of Fig. 5 we see that improvements are almost always positive, up to a maximum of 8% in 80-m wind speed MAE and 10% in 80-m wind power MAE. The impact on 80-m wind power is larger because the power increases approximately as the cubic power of the wind speed in the range of speeds between 5-12 m s$^{-1}$ (International Electrotechnical Commission, 2007).



## 4.2 Impact of model finer horizontal grid spacing (HRRRNEST vs HRRR)

Improvements due to finer horizontal grid spacing are larger than those due to the experimental physics. The impact of the finer horizontal grid spacing in the control runs (HRRRNEST CNT vs HRRR CNT) is shown in Fig. 6 for 80-m wind speed (left panels) and 80-m wind power (right panels). MAE values in the upper panels are in red for the HRRR CNT runs and in yellow for the HRRRNEST CNT. In the bottom panels of Fig. 6 we see a large percentage improvement in MAE due to finer horizontal grid spacing, particularly at nighttime and during the morning transition (approximately between 0100 UTC and 1500 UTC). Improvements due to finer horizontal grid spacing are larger than those due to the experimental physics in Fig. 5, with values now up to 10% in 80-m wind speed MAE and up to 15% in 80-m wind power MAE. The percentage improvements are smaller during daytime, when the model with larger horizontal grid spacing had lower MAE compared to nighttime.

In Fig. 7 we compare the improvements in 80-m wind speed MAE due to the experimental physics (left panels) from the HRRR (shown previously in Fig. 5) with those found in the HRRRNEST, and the improvements due to finer horizontal grid spacing (right panels) from the CNT simulations (shown previously in Fig. 6) with those found in the EXP simulations. The dark blue curve shows the impact of the experimental physics on the models with larger horizontal grid spacing (HRRR EXP vs HRRR CNT), while light blue shows the impact of the experimental physics on the models with finer horizontal grid spacing (HRRRNEST EXP vs HRRRNEST CNT). The red curve shows the impact of finer horizontal grid spacing on the CNT runs (HRRRNEST CNT vs HRRR CNT),while the impact of finer horizontal grid spacing on the EXP runs (HRRRNEST EXP vs HRRR EXP) is shown in orange. When averaged over the four reforecast periods, the impact of the experimental physics (left upper panel) is quite similar between the higher and finer horizontal grid spacing models, however when considering the four reforecast periods separately (lower left smaller panels) the impact varies considerably. For example, in summer the impact of the experimental physics on the HRRRNEST is mostly neutral (light blue curve), while in the HRRR it is actually producing a negative impact (dark blue curve). In contrast, while the impact of the experimental physics is positive for both horizontal grid spacings in winter, it is very positive for the HRRR (dark blue curve). This variation could be due to changes in the physics that are grid-spacing dependent, making the impact different for HRRR and HRRRNEST. Similar considerations can be made for the improvement due to finer horizontal grid spacing (right panels). When averaged over the four reforecast periods (right upper panel) the impact of the finer horizontal grid spacing is similar between the models with different physics. However, for the winter reforecast period (lower right panel) the impact of the finer horizontal grid spacing on the EXP runs is mostly neutral (orange curve), while for the CNT runs it is clearly positive (red curve).

## 4.3 Impact to the statistics due to the experimental physics and finer horizontal grid spacing (HRRRNEST EXP vs HRRR CNT)

As a final step of the analysis, the combined impact on 80-m wind speed MAE of the experimental physics and finer horizontal grid spacing, comparing the HRRRNEST EXP to HRRR CNT is shown in Fig. 8. Consistent with the results presented in the previous sections, we find that the combination of the experimental physics and finer horizontal grid spacing produces even larger improvements, always positive and up to a maximum of 14% in the 80-m wind speed MAE (lowest left panel) and up



to a maximum of 18% in 80-m wind power MAE (lowest right panel). Again, larger improvements are found during nighttime and during the morning transition, with smaller improvement found during daytime when the models had lower MAEs.

To condense the results presented in this section, a summary plot with the percentage improvements on MAE due to the experimental physics, finer horizontal grid spacing, and the combination of the two, for the four reforecast periods separately and averaged together is presented in Fig. 9 (left panel is for 80-m wind speed MAE and right panel is for 80-m wind power MAE results). For this plot the results are averaged over all sites, between the two initialization times, and over all reforecast horizons between 03 and 24. Averaged over the four reforecast periods (bars on the right side of each panel) we see positive improvements due to the experimental physics in the HRRR (in dark blue) and HRRRNEST (in light blue) reforecast runs, up to ~3% in terms of 80-m wind speed MAE and ~4% in terms of 80-m wind power MAE. Finer horizontal grid spacing in the CNT (in red) and EXP (in orange) reforecast runs produces improvements of up to ~5% for 80-m wind speed MAE and ~7% for 80-m wind power MAE. In grey is the improvement due to the combination of the experimental physics and finer horizontal grid spacing (HRRRNEST EXP vs HRRR CNT), approximately 7% for 80-m wind speed MAE and ~11-12% for 80-m wind power MAE. Considering the individual reforecast periods, in winter the improvements due to the experimental physics are very large for the HRRR, as are those due to the combination of the experimental physics and finer horizontal grid spacing (13% for 80-m wind speed MAE and 21% for 80-m wind power MAE). Negative impacts to the improvement due to the changes in the physics of the HRRR (dark blue bars) are found in spring and summer, down to ~-7% for 80-m wind speed MAE and ~-10% for 80-m wind power MAE. What causes the dark blue bar in summer 2016 to be so negative? To answer this question, in the next section we investigate the improvements as a function of the different meteorological phenomena characteristic of this area (cold pools, gap flows, easterly flows, mountain waves, topographic wakes, and convective outflows).

## 4.4 Statistical results as a function of the different meteorological phenomena

The improvements due to the experimental physics and finer horizontal grid spacing (and to the combination of the two) as a function of the different meteorological phenomena common to this area are presented in Fig. 10. For this analysis we take advantage of the WFIP2 Event Log, which was created and updated regularly during WFIP2 by several meteorologists documenting the meteorological conditions of relevance in the area and is available on the DAP (Shaw et al., 2019). The WFIP2 meteorologists based their classification of events on WFIP2 observations and other surface observations, real-time and global model forecasts, satellite images, and local radio-soundings. In the Event Log document, days and characteristics of the different meteorological phenomena were recorded, with the possibility that on some days multiple phenomena could occur at the same time. Although the categorization of the days into different meteorological phenomena involves a certain level of subjectivity, the final classification generally had the concurrence of several meteorologists, and so we believe the results are consistent and robust. For the plot in Fig. 10 the results are averaged over all sites, between the two initialization times, over all reforecast horizons between 03 and 24 and over the four reforecast periods. The number of days over which each specific phenomenon takes place is in the parentheses on the x-axis label. On the far right are the improvements averaged (weighted by the number of cases) over all the different phenomena. Since on some days multiple phenomena might occur at





the same time, same days can be counted multiple times in the average, which consequently is not exactly the same as that in Fig. 9. From this analysis there is no improvement in the 80-m wind speed MAE due to the modifications in the physics of the HRRR (in blue) for mountain waves and topographic wakes, while for the other meteorological phenomena the impact due to the experimental physics is positive. In truth, this figure does not tell the entire story.

As shown in Fig. 10, the number of days with gap flow events is very high (145), and if we plot the same figure separately for each of the four reforecast periods (Fig. 11), we see that the gap flow events are almost equally distributed over the four reforecast periods (34 in spring 2016, upper left panel; 41 in summer 2016, upper right panel; 38 in fall 2016, lower left panel; and 32 in winter 2017, lower right panel). Mountain wave (54 days in total) and topographic wave events (30 days in total) are also distributed over all reforecast periods. From Fig. 11 we can say that the impact of the experimental physics and finer

horizontal grid spacing on 80-m wind speed MAE during gap flow, mountain waves and topographic wakes situations differs from season to season (negative in spring and summer and positive for fall and winter).

Consequently, the negative blue bar in spring and summer visible in Fig. 9 is not only due to negative impact of mountain wave and topographic wake days, but also to gap flow days in spring and summer (upper right and lower left panels of Fig. 11). From Fig. 11 we also note that easterly flow is a category with a more consistent impact, always being improved by the

15 experimental HRRR physics. Cold pool events are also consistently improved by the experimental HRRR physics; this type of event happens mostly in fall and winter (only one event is found in spring, therefore its impact cannot be considered statistically significant).

To better understand the reasons for the lack of MAE improvement in the HRRR EXP vs HRRR CNT runs during gap flow days in summer, in Fig. 12 we present the aggregated time series of 80-m wind speed MAE (upper panel) and wind speed

(lower panel) for the 22 sites for part of the summer reforecast period (all of the summer reforecast period shows a similar behaviour). HRRR CNT is shown in red, HRRR EXP is in blue, and observations are in black. In the lower panel, gap flow days are highlighted with the red shaded areas. From the time series in the upper panel of Fig. 12, we see that the 80-m wind speed MAE of the HRRR EXP (blue line) is often larger than that of the HRRR CNT (red line). For almost all of the gap flow days the HRRR EXP forecasts the down-ramp too early, compared to the observations and to the HRRR CNT. Similar results

were found for the spring reforecast period (not shown).

Although from Fig. 11 we see the experimental physics generally improves the HRRR during cold pool events, we next examine details of the when and how this improvement occurs. Fig. 13 is similar to Fig. 12, but for part of the winter reforecast period. In the lower panel days identified in the Event Log as experiencing cold pools are highlighted with the blue shaded areas. In the time series shown in the upper panel of Fig. 13, a period when the 80-m wind speed MAE of the HRRR EXP

(blue line) is larger than the HRRR CNT (red line) is highlighted with the red oval, while at a later time (inside the blue oval) the opposite is true. Differences between these cold pool events were examined using the WFIP2 real-time model observation evaluation website (http://wfip.esrl.noaa.gov/psd/programs/wfip2/). This website was used through the duration of the WFIP2 field campaign for daily monitoring of model forecasts and instrument health (Wilczak et al., 2019a).





Time-height cross sections of microwave radiometer temperature (upper panels), and winds from the radar wind profiler superimposed on radio acoustic sounding system virtual temperature (lower panels) at Wasco, OR, for January 4, 2017 (left) and January 19, 2017 (right) are presented in Fig. 14. From the cross sections of both instruments we see that the cold pool at the beginning of January (left panels) is brought in by sustained easterly winds and has weaker stable stratification compared

to the cold pool event in the second half of January (right panels), which is characterized by very low wind speeds close to the surface and more strongly stable stratification. Thus, although these periods are both listed as cold pool events, they have different atmospheric characteristics. In the first case the experimental physics in the HRRR EXP run does not help the model to outperform the HRRR CNT, while in the second case it does. A large wind speed deficit in the HRRR EXP forecast on January, 4, 2017 (visible in the red oval in the lower panel of Fig. 13) might occur because the HRRR EXP model has too

much drag due to the SGS and/or because of the wind farm parameterization, with wind farms just upwind, east of Wasco. In contrast, in January 18, 2017 a large wind speed excess in the HRRR CNT forecast (visible in the blue oval in the lower panel of Fig. 13) occurs because of 1) not enough drag in the HRRRR CNT to reduce the strong winds immediately above the cold pool, 2) too much mixing at the top of the cold pool, which may be due to too large mixing lengths, and 3) to "horizontal" mixing along sloped sigma coordinates, which contribute to vertical mixing. Given the very different wind and stability profiles

characteristics of the two cold pool events, having routinely available observations of these profiles and assimilating them into the models would likely improve their short- term forecast skill. The need of a network of ground-based profiling instruments to improve numerical weather prediction and operational forecasting is also strongly advocated by the National Research Council (2009).

### 4.5 Bias correction impact on the improvements

Next, we evaluate whether the improvements measured in the previous sections are mainly due to reducing the biases of the models (the systematic component of the error) or if the model improvements also address the random component of error. To this aim the model 80-m wind speed output needs to be bias corrected before the bulk statistics and the relative improvements can be computed. Several methods have been investigated in the literature to remove the systematic component of the error from model outputs. For this study we think that, due to the nature of the 80-m wind speed biases presented in Fig. 2, at least

two possible bias correction methods have to be considered. The first one would be to remove the mean bias from each model, at each site, and for each reforecast period separately ("mean bias"). The second method would be to remove the mean bias from each model, at each site, for each of the reforecast periods and for each of the hour of the day separately ("diurnal bias"). Since, as it is clear from Fig. 2, the nature of the bias differs among the models, we examine the impacts of both of these simple bias correction methods. In Fig. 15 we present the same results presented in Fig. 9, but after applying the "mean bias" correction

(upper panel) and the "diurnal bias" correction (lower panel).

The "mean bias" correction enhances the improvement due to the experimental physics in the HRRR and HRRRNEST models so that it is positive for all reforecast periods (blue and light blue bars, comparing Fig. 15 to Fig. 9). This improvement indicates that the experimental physics improves the random component of the model error, even if the experimental physics might





degrade the systematic component: the right panel of Fig. 4 shows that the bias of the HRRR EXP model is larger than the bias of the HRRR CTL model. In comparison, applying the "diurnal bias" correction also increases the improvement due to the experimental physics (dark blue and light blue bars) over all reforecast periods and for their average, while the improvements due to finer horizontal grid spacing in the models (red and orange bars) actually decrease.

**5 Summary and conclusions**

Measurements collected by 19 sodars and 3 lidars during the second Wind Forecast Improvement Project (WFIP2), an 18-month field campaign in the Columbia River Gorge and Basin area, were used to validate model runs by the High Resolution Rapid Refresh (HRRR) model (3 km horizontal grid spacing) and its nested version (HRRRNEST, 750 m horizontal grid spacing).

The models were run for four 6-week reforecast periods (one for each season) in control (CNT) and experimental (EXP) configurations, where the EXP runs included new parameterizations to the HRRR and HRRRNEST physics suites (i.e. representation of wind farms and of drag associated with subgrid-scale (SGS) topography in the HRRR), improvements to existing parameterizations (i.e. boundary-layer and surface-layer schemes, cloud–radiation interaction), and improvements to numerical methods (i.e. finite differencing of the horizontal diffusion). Results showed that:

15 - 80-m wind speed MAE and bias vary significantly through the diurnal cycle, with time of day being more important at determining the 80-m wind speed MAE and bias values than either the initialization time or the forecast horizon.

- The HRRR EXP reforecast run reduces the diurnal trend in the bias, but results in a near constant negative bias, possibly by exaggerating the drag due to sub-grid scale orography added to the HRRR physics suite (but not to the HRRRNEST).

- The 80-m wind speed biases have lower values (often negative) at lower elevations, but increase with the site elevation. Differences in the sub-grid scale terrain inhomogeneity did not help explain any of the bias or MAE in the results.

- The experimental physics in the HRRR reduces 80-m wind speed MAE by 3-4% and 80-m wind power MAE by 4-5 %.

- Finer model horizontal grid spacing improves 80-m wind speed MAE in the control runs, particularly at nighttime and during the morning transition. Smaller improvements occur during daytime, when the larger horizontal grid spacing model had lower MAE than at nighttime. The finer horizontal grid spacing of the HRRRNEST produces average improvement values up to 5% in 80-m wind speed MAE and up to 7-8% in 80-m wind power MAE.

- The combined impact on 80-m wind speed MAE of the experimental physics and finer horizontal grid spacing
produces an even larger reduction in MAE, averaging 7-8% for 80-m wind speed and 11-12% for 80-m wind power.





- Improvements due to the experimental physics and finer horizontal grid spacing depend on season but almost always are positive. However, in spring and summer, the experimental physics in the HRRR runs increases the 80-m wind speed MAE.

- The negative impact of the experimental physics on the HRRR found in spring and summer results from degradation of the HRRR EXP on days experiencing gap flows, mountain waves and topographic wakes, and is probably due to the representation of drag in the HRRR EXP. In particular, for almost all of the summer gap flow days, the HRRR EXP predicts the down-ramps occurring at the end of the events too early.

- Although cold pool forecast skill improves due to the experimental physics in the models, different types of cold pools are predicted with varying skill. If routinely available observations of wind and stability profiles were
10 assimilated into the models, short term forecast skill would likely improve.

- "Mean bias" and "diurnal bias" corrections of the 80-m wind speed model outputs demonstrated that the experimental physics improves the random component of the model errors. The impacts of the different bias corrections on the improvements due to finer horizontal grid spacing in the models are mixed.

The strength of WFIP2 came from many observational scientists and model developers working closely together, steering the observational-based process understanding to guide model improvements which were later transitioned into operations. The current analysis quantifies the skill added by improvements made to the models within four months towards the end of WFIP2. A model freeze was then imposed so that the models could be run in EXP and CNT configurations over the four chosen reforecast periods. Further improvements to the models, based on WFIP2 observations, continue to be made which will become
part of the operational HRRR in the near future.

**Authors' contribution**

Laura Bianco, Irina V. Djalalova, and James M. Wilczak contributed with the data preparation, main analysis and organization of the results in the paper. Joseph B. Olson and Jaymes S. Kenyon worked at the improvements of the HRRR and HRRRNEST parameterizations, ran the models in CNT and EXP configurations, and contributed with useful discussion to improve the
25 manuscript. Aditya Choukulkar contributed with the categorization of the atmospheric phenomena in the Event Log, with observational data, and with useful discussion to improve the manuscript. Larry K. Berg, Harindra J. S. Fernando, Eric P. Grimit, Raghavendra Krishnamurthy, Julie K. Lundquist, Paytsar Muradyan, Mikhail Pekour, Yelena Pichugina, Mark T. Stoelinga, and David D. Turner contributed with observational data and with useful discussion to improve the manuscript.




**Data and code availability**

The operational HRRR model is not entirely open source (data assimilation/cycling scripts/etc), but updates to the model parameterizations used in the HRRR are deposited periodically to the official repository for the Advanced Research version of the Weather Research and Forecasting (WRF-ARW) model, maintained by the National Center for Atmospheric Research

(NCAR), which is open source (https://github.com/wrf-model/WRF). A branch from this repository was created for WFIP2 testing, based on WRF-ARWv3.9. This branch is currently stored at https://github.com/joeolson42/WFIP2. This branch is no longer under development and all improvements have been transferred to NCAR's official repository.

Details on the improvements applied to the HRRR and HRRRNEST parameterizations can be also found in Olson et al. (2019a).

All dataset used in this study are freely available to the public from the DOE Data Archive and Portal (DAP; https://a2e.energy.gov/projects/wfip2).

Please contact the corresponding author for additional details, if needed.

**Acknowledgements**

We thank all the people involved in WFIP2 for site selection, leases, instrument deployment and maintenance, data collection,

and data quality control. Funding for this work was provided by the DOE, Office of Energy Efficiency and Renewable Energy, Wind Energy Technologies Office, and by the NOAA/ESRL Atmospheric Science for Renewable Energy program. This work was authored (in part) by NREL, operated by the Alliance for Sustainable Energy, LLC, for the U.S. DOE, under Contract No. DE-AC36-08GO28308, with funding provided by the U.S. DOE Office of Energy Efficiency and Renewable Energy Wind Energy Technologies. Pacific Northwest National Laboratory is operated by Battelle Memorial Institute for the U.S. DOE

under Contract No. DE-AC05-76RL01830.





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

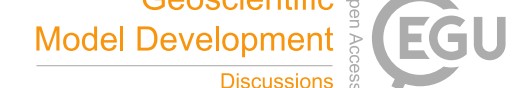



| Type of instr. | Site ident. name | Lat (N) | Lon (W) | Alt (m asl) | Terrain complexity std (m) | Data availability (%) | Institution in charge |
|---|---|---|---|---|---|---|---|
| sodar | AON1 | 45.505 | 119.491 | 706 | 64 | Spr 16: 96<br>Sum 16: 96<br>Fall 16: 91<br>Win 17: 33 | Vaisala |
| sodar | AON2 | 45.554 | 120.156 | 356 | 13 | Spr 16: 98<br>Sum 16: 98<br>Fall 16: 93<br>Win 17: 94 | Vaisala |
| sodar | AON3 | 45.938 | 119.406 | 116 | 12 | Spr 16: 97<br>Sum 16: 98<br>Fall 16: 92<br>Win 17: 84 | Vaisala |
| sodar | AON4 | 45.637 | 120.680 | 432 | 34 | Spr 16: 98<br>Sum 16: 97<br>Fall 16: 92<br>Win 17: 72 | Vaisala |
| sodar | AON5 | 45.575 | 120.747 | 456 | 13 | Spr 16: 99<br>Sum 16: 99<br>Fall 16: 93<br>Win 17: 95 | Vaisala |
| sodar | AON6 | 45.516 | 120.781 | 731 | 81 | Spr 16: 97<br>Sum 16: 84<br>Fall 16: 82<br>Win 17: 89 | Vaisala |
| sodar | AON7 | 45.631 | 121.069 | 166 | 55 | Spr 16: 97<br>Sum 16: 16<br>Fall 16: 0<br>Win 17: 86 | Vaisala |
| sodar | AON8 | 45.602 | 121.589 | 703 | 98 | Spr 16: 34<br>Sum 16: 0<br>Fall 16: 0<br>Win 17: 0 | Vaisala |
| sodar | AON9 | 45.374 | 121.330 | 836 | 57 | Spr 16: 0<br>Sum 16: 0<br>Fall 16: 0<br>Win 17: 51 | Vaisala |
| sodar | BOR | 45.816 | 119.812 | 112 | 6 | Spr 16: 95 | NOAA/ARL |





| | | | | | | Sum 16: 96<br>Fall 16: 74<br>Win 17: 83 | |
|---|---|---|---|---|---|---|---|
| sodar | CDN | 45.245 | 120.169 | 891 | 25 | Spr 16: 8<br>Sum 16: 37<br>Fall 16: 84<br>Win 17: 97 | DOE/NREL |
| sodar | DCR | 45.165 | 120.656 | 795 | 26 | Spr 16: 96<br>Sum 16: 98<br>Fall 16: 97<br>Win 17: 92 | DOE/NREL |
| sodar | GDL | 45.805 | 120.849 | 501 | 16 | Spr 16: 95<br>Sum 16: 98<br>Fall 16: 90<br>Win 17: 87 | DOE/ANL |
| sodar | PVE | 44.285 | 120.901 | 991 | 42 | Spr 16: 96<br>Sum 16: 96<br>Fall 16: 92<br>Win 17: 57 | NOAA/ARL |
| sodar | RFS | 45.691 | 120.746 | 62 | 80 | Spr 16: 48<br>Sum 16: 4<br>Fall 16: 11<br>Win 17: 23 | UND |
| sodar | RTK | 45.364 | 120.747 | 708 | 19 | Spr 16: 94<br>Sum 16: 98<br>Fall 16: 89<br>Win 17: 41 | DOE/PNNL |
| sodar | WCO | 45.590 | 120.672 | 462 | 25 | Spr 16: 81<br>Sum 16: 88<br>Fall 16: 69<br>Win 17: 71 | NOAA/ARL |
| sodar | WWL | 46.095 | 118.261 | 382 | 34 | Spr 16: 91<br>Sum 16: 85<br>Fall 16: 83<br>Win 17: 97 | DOE/ANL |
| sodar | YKM | 46.572 | 120.551 | 330 | 19 | Spr 16: 96<br>Sum 16: 73<br>Fall 16: 25<br>Win 17: 85 | DOE/ANL |
| scanning | ARL | 45.720 | 120.187 | 266 | 56 | Spr 16: 100 | NOAA/ESRL |





| lidar | | | | | | Sum 16: 100<br>Fall 16: 28<br>Win 17: 95 | |
|---|---|---|---|---|---|---|---|
| profiling lidar | GDR | 45.516 | 120.780 | 725 | 81 | Spr 16: 90<br>Sum 16: 90<br>Fall 16: 71<br>Win 17: 0 | CU |
| profiling lidar | VCR | 45.954 | 118.688 | 542 | 69 | Spr 16: 93<br>Sum 16: 97<br>Fall 16: 78<br>Win 17: 45 | LLNL |

**Table 1: List of the instruments used in this study with site identification name, latitude, longitude, elevation, terrain complexity, percentage of data availability, and institution in charge.**





**Figure captions**

**Figure 1:** Diurnally averaged 80-m wind speed MAEs for: HRRR CNT (red curves), HRRR EXP (blue curves), HRRRNEST CNT (yellow curves), and HRRRNEST EXP (black curves). Panel a) shows the MAEs averaged over the four reforecast periods, panel b) are MAEs for the spring 2016 reforecast period, c) for summer 2016, d) for fall 2016 and e) for winter 2017. Initialization times at 0000UTC (Z00) are represented with O's and at 1200UTC (Z12) with X's, while the solid bold lines are the averages between the Z00 and Z12 values. Red and blue arrows on the y-axes represent the sunrise and sunset times, respectively.

**Figure 2:** As in Figure 1 but for the 80-m wind speed biases.

**Figure 3:** 80-m wind speed MAEs (on the left) and biases (on the right) averaged over the four reforecast periods. Initialization times are represented with the O's (Z00 runs) and with the X's (Z12 runs), while the solid bold lines are the averages between the Z00 and Z12 values.

**Figure 4:** 80-m wind speed bias (model-observations) normalized by the averaged (observed) 80-m wind speed at each site for the four reforecast runs as a function of site elevation for the four reforecast periods separately: panel a) is for the spring 2016 reforecast period, b) for summer 2016, c) for fall 2016 and d) for winter 2017). Sites are sorted from low to high elevation (from Rufus at 62 m asl to Prineville at 991 m asl).

**Figure 5:** Left panels: HRRR EXP vs HRRR CNT MAE for 80-m wind speed. Right panels: As on the left, but for 80-m wind power, showing the impact of the experimental physics. Upper panels are MAEs, middle panels are differences between MAEs of the HRRR CNT run and HRRR EXP run (error bars represent the $\pm 1.96\sigma/\sqrt{n}$ interval of the 95% confidence level), and lower panels are the percentage MAE relative improvement of the HRRR EXP model over the HRRR CNT model.

**Figure 6:** As in Fig. 5 but for HRRRNEST CNT (in yellow) vs HRRR CNT (in red) runs, showing impact of finer model horizontal grid spacing.

**Figure 7:** Improvements in 80-m wind speed MAE due to the experimental physics (left panels) and finer horizontal grid spacing (right panels) for the four reforecast periods averaged together (upper panels) and for the four reforecast period separately (lower smaller panels) for all reforecast runs. In dark blue is HRRR EXP vs HRRR CNT, in light blue HRRRNEST EXP vs HRRRNEST CNT, in red is HRRRNEST CNT vs HRRR CNT, and in orange HRRRNEST EXP vs HRRR EXP. Red and blue arrows on the y-axes represent the sunrise and sunset times, respectively.

**Figure 8:** As in Fig. 6 but for HRRRNEST EXP (in black) vs HRRR CNT (in red) runs.

**Figure 9:** Left panel: percentage improvements on 80-m wind speed MAE due to the experimental physics, finer horizontal grid spacing, and the combination of the two, for the four reforecast periods separately and averaged together. Right panel: Same as on the left, but for 80-m wind power MAE results.

**Figure 10:** Improvements due to the experimental physics (blue and light blue), finer horizontal grid spacing (red and orange), and to the combination of the two (gray) as a function of the different meteorological phenomena common to the WFIP2 area.

**Figure 11:** Same as in Fig. 10, but for the four reforecast periods individually (spring, upper left panel; summer, upper right panel; fall, lower left panel; and winter, lower right panel).

**Figure 12:** Time series of 80-m wind speed MAE (upper panel) and 80-m wind speed (lower panel) for the summer reforecast period. HRRR CNT is in red, HRRR EXP is in blue, observations are in black. In the lower panel days identified in the Event Log as experiencing gap flows are highlighted with the red shaded areas.

**Figure 13:** As in Fig. 12, but for part of the winter 2017 reforecast period.

**Figure 14:** Time-height cross sections of microwave radiometer temperature (upper panels), and radio acoustic sounding system virtual temperature and winds from the radar wind profiler (lower panels) at Wasco, OR, for January 4, 2017 (left) and January 19, 2017 (right). Red and blue arrows on the y-axes represent the sunrise and sunset times, respectively.

**Fig. 15.** Percentage improvements on 80-m wind speed MAE (after bias correcting the model output) due to the experimental physics, finer horizontal grid spacing, and the combination of the two, for the four reforecast periods separately and averaged together. Upper panel: results using a "mean bias" correction; lower panel: results using a "diurnal bias" correction.



**Figure 1: Diurnally averaged 80-m wind speed MAEs for: HRRR CNT (red curves), HRRR EXP (blue curves), HRRRNEST CNT (yellow curves), and HRRRNEST EXP (black curves). Panel a) shows the MAEs averaged over the four reforecast periods, panel b) are MAEs for the spring 2016 reforecast period, c) for summer 2016, d) for fall 2016 and e) for winter 2017. Initialization times at 0000UTC (Z00) are represented with O's and at 1200UTC (Z12) with X's, while the solid bold lines are the averages between the Z00 and Z12 values. Red and blue arrows on the y-axes represent the sunrise and sunset times, respectively.**





**Figure 2: As in Figure 1 but for the 80-m wind speed biases.**





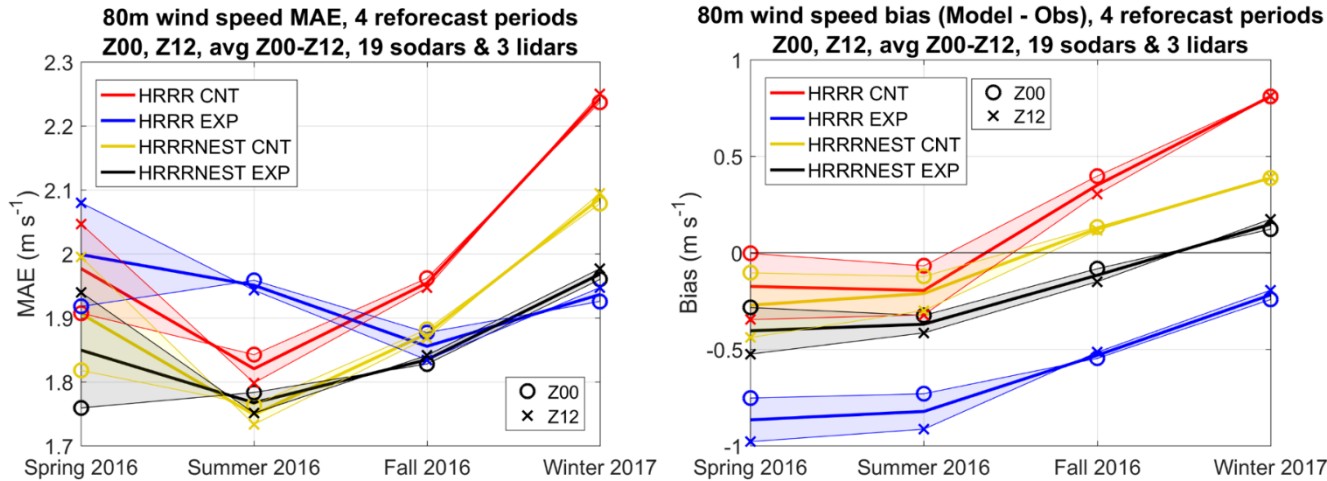

**Figure 3: 80-m wind speed MAEs (on the left) and biases (on the right) averaged over the four reforecast periods. Initialization times are represented with the O's (Z00 runs) and with the X's (Z12 runs), while the solid bold lines are the averages between the Z00 and Z12 values.**





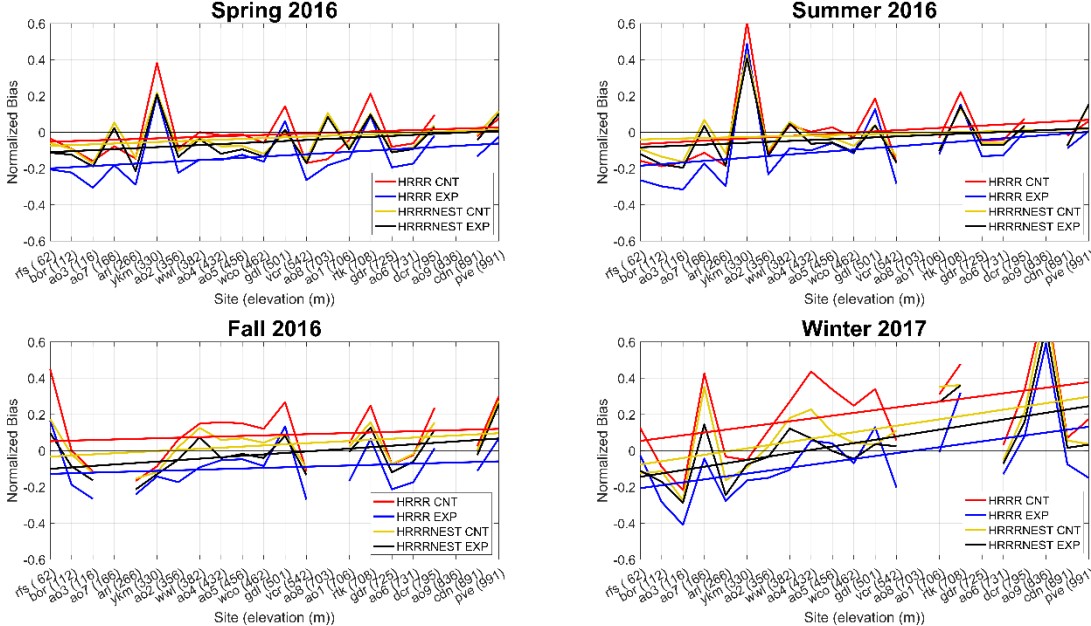

**Figure 4: 80-m wind speed bias (model-observations) normalized by the averaged (observed) 80-m wind speed at each site for the four reforecast runs as a function of site elevation for the four reforecast periods separately: panel a) is for the spring 2016 reforecast period, b) for summer 2016, c) for fall 2016 and d) for winter 2017). Sites are sorted from low to high elevation (from Rufus at 62 m asl to Prineville at 991 m asl).**





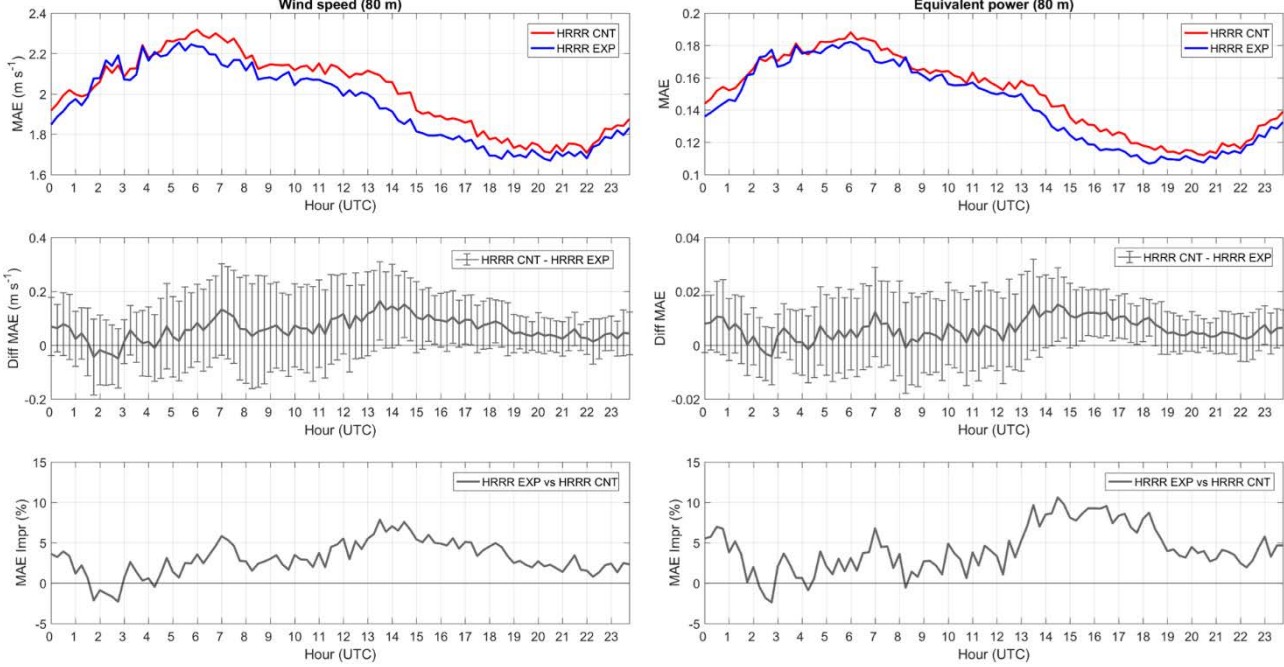

**Figure 5: Left panels: HRRR EXP vs HRRR CNT MAE for 80-m wind speed. Right panels: As on the left, but for 80-m wind power, showing the impact of the experimental physics. Upper panels are MAEs, middle panels are differences between MAEs of the HRRR CNT run and HRRR EXP run (error bars represent the ±1.96σ/√‾n interval of the 95% confidence level), and lower panels are the percentage MAE relative improvement of the HRRR EXP model over the HRRR CNT model.**





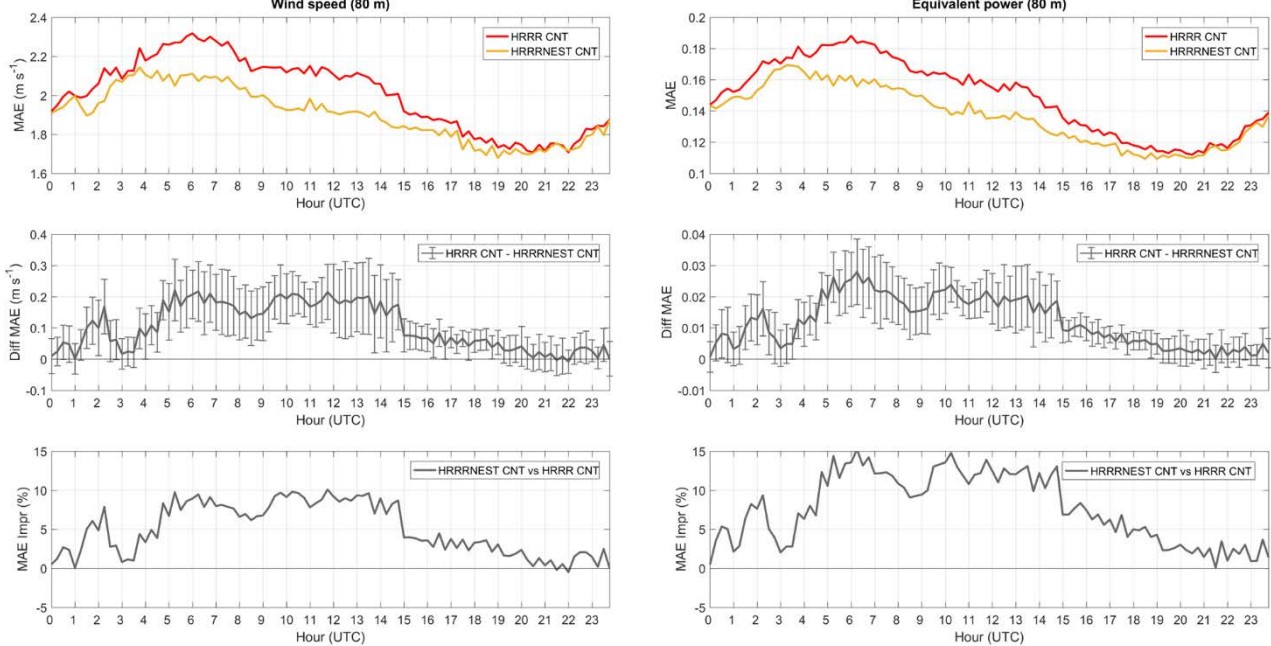

**Figure 6: As in Fig. 5 but for HRRRNEST CNT (in yellow) vs HRRR CNT (in red) runs, showing impact of finer model horizontal grid spacing.**





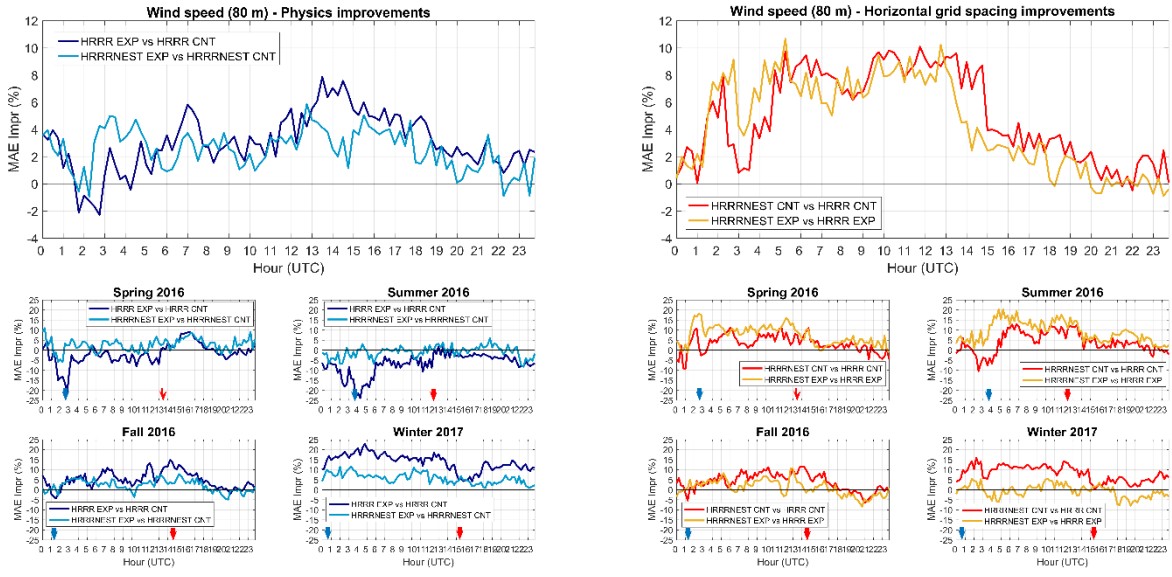

**Figure 7: Improvements in 80-m wind speed MAE due to the experimental physics (left panels) and finer horizontal grid spacing (right panels) for the four reforecast periods averaged together (upper panels) and for the four reforecast period separately (lower smaller panels) for all reforecast runs. In dark blue is HRRR EXP vs HRRR CNT, in light blue HRRRNEST EXP vs HRRRNEST CNT, in red is HRRRNEST CNT vs HRRR CNT, and in orange HRRRNEST EXP vs HRRR EXP. Red and blue arrows on the y-axes represent the sunrise and sunset times, respectively.**



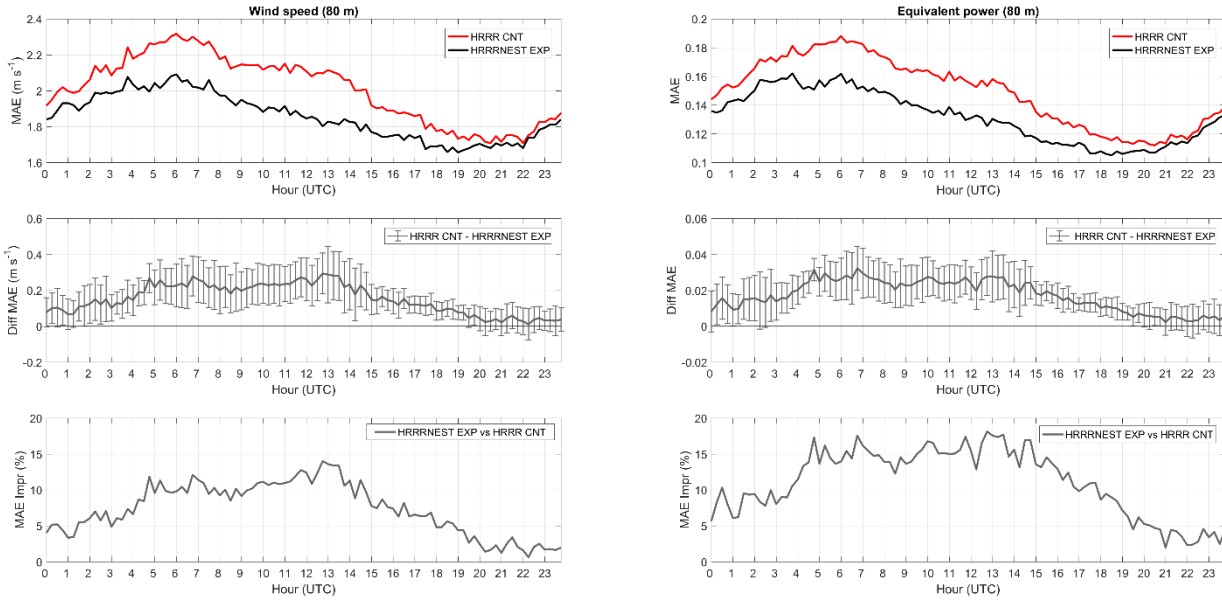

**Figure 8: As in Fig. 6 but for HRRRNEST EXP (in black) vs HRRR CNT (in red) runs.**




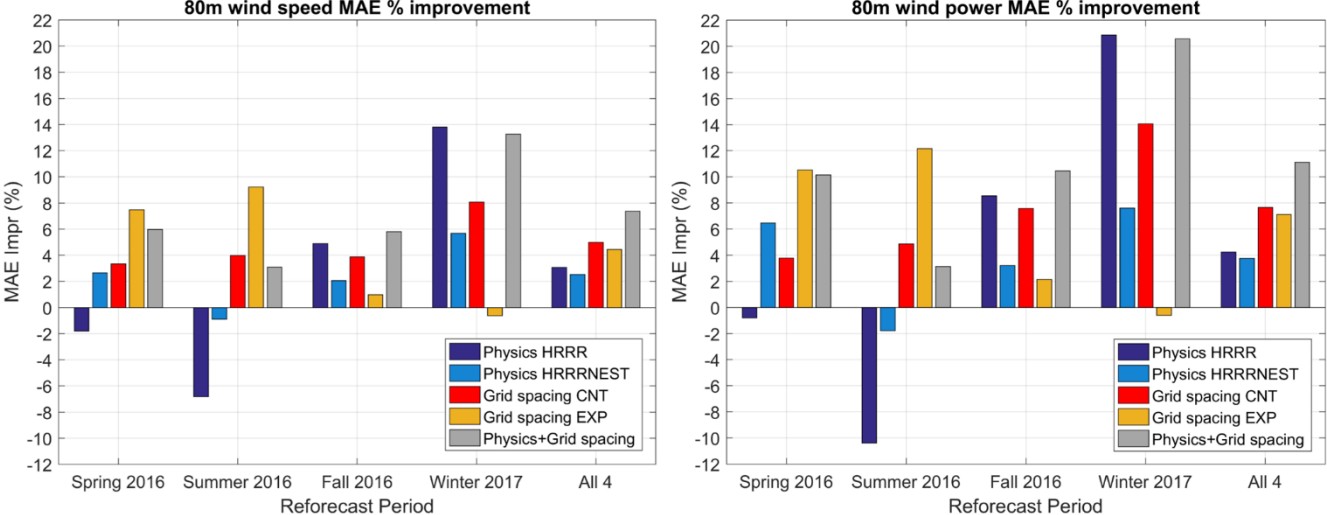

**Figure 9: Left panel: percentage improvements on 80-m wind speed MAE due to the experimental physics, finer horizontal grid spacing, and the combination of the two, for the four reforecast periods separately and averaged together. Right panel: Same as on the left, but for 80-m wind power MAE results.**





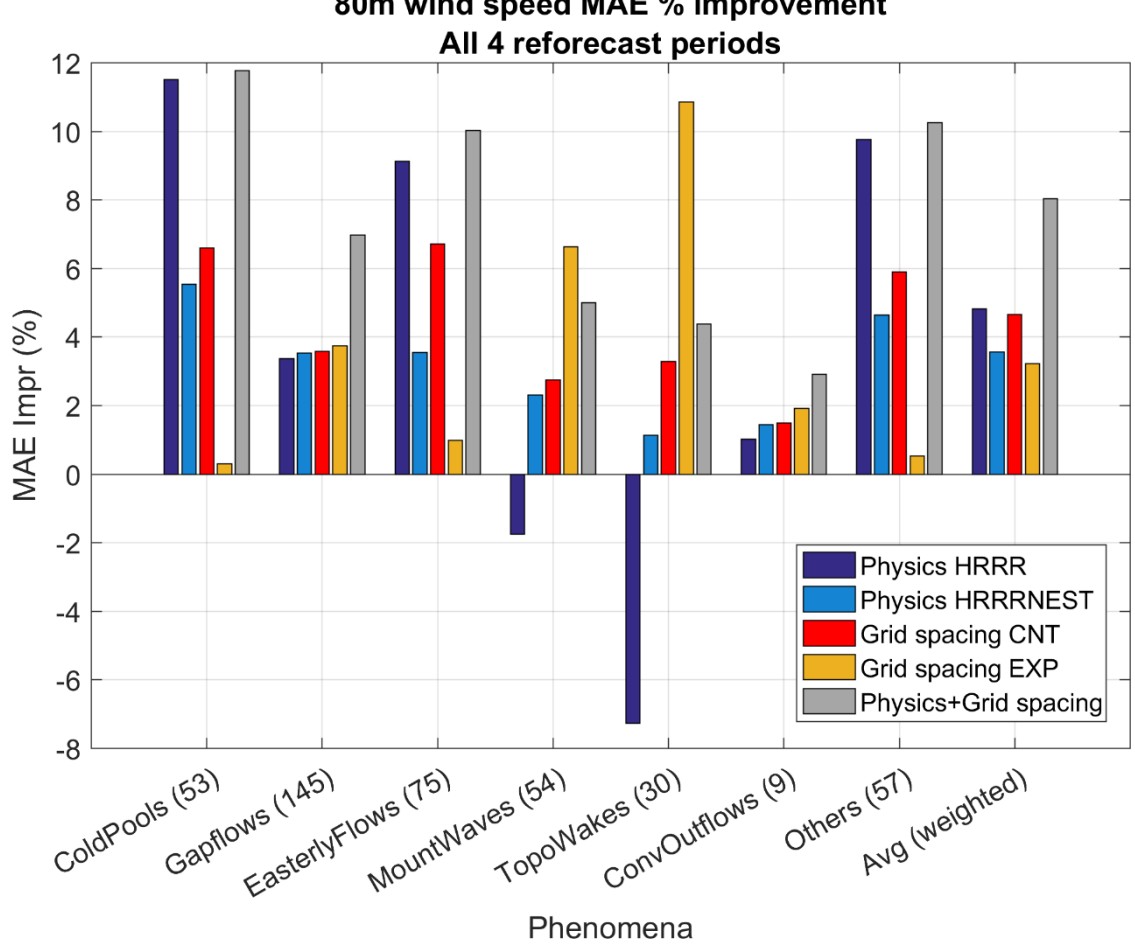

**Figure 10: Improvements due to the experimental physics (blue and light blue), finer horizontal grid spacing (red and orange), and to the combination of the two (grey) as a function of the different meteorological phenomena common to the WFIP2 area.**

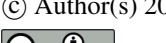



**Figure 11: Same as in Fig. 10, but for the four reforecast periods individually (spring, upper left panel; summer, upper right panel; fall, lower left panel; and winter, lower right panel).**



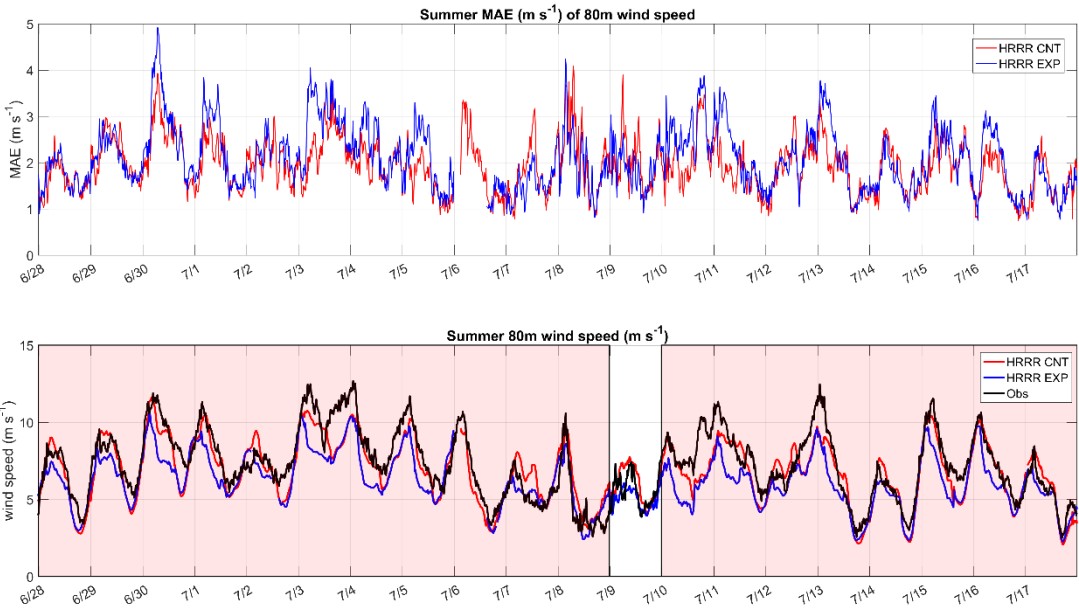

**Figure 12: Time series of 80-m wind speed MAE (upper panel) and 80-m wind speed (lower panel) for the summer reforecast period. HRRR CNT is in red, HRRR EXP is in blue, observations are in black. In the lower panel days identified in the Event Log as experiencing gap flows are highlighted with the red shaded areas.**



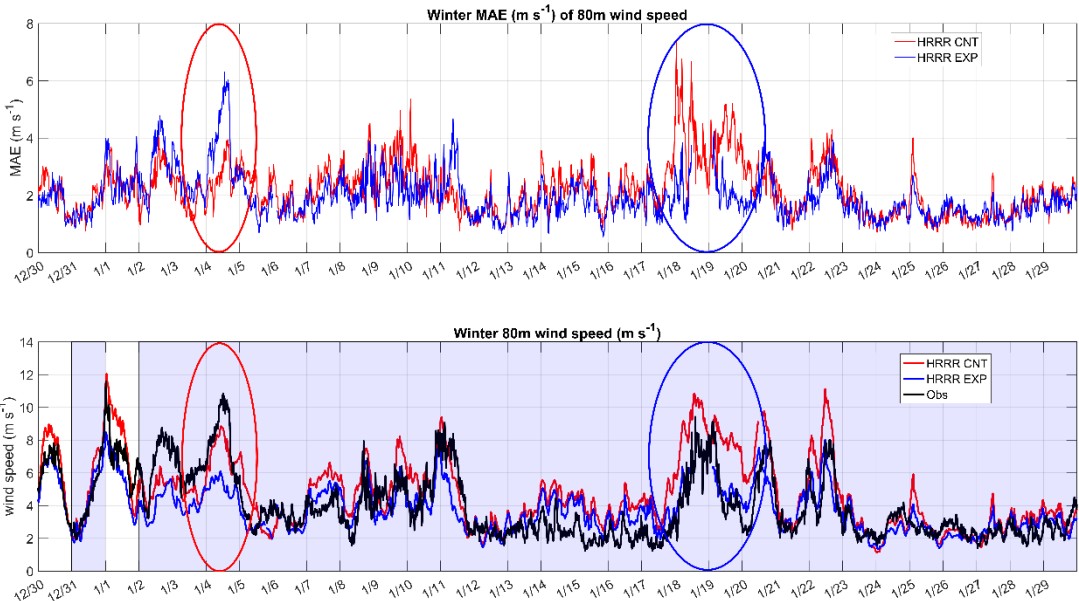

**Figure 13: As in Fig. 12, but for part of the winter 2017 reforecast period.**





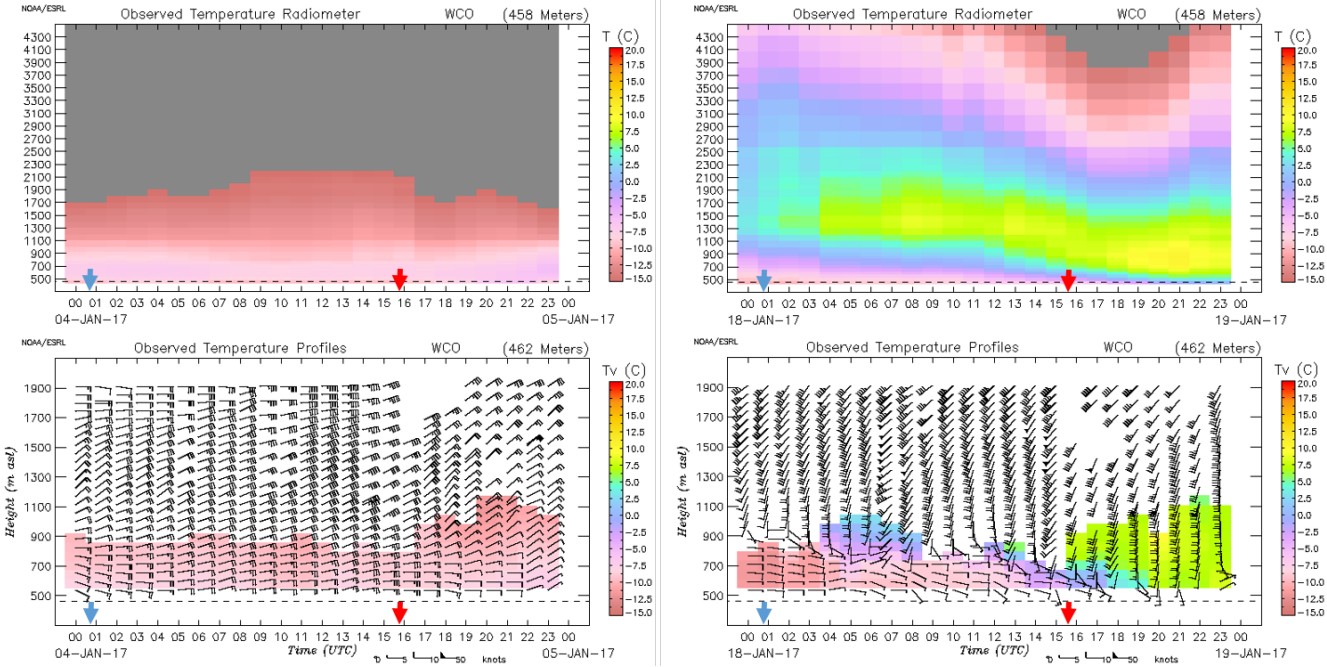

**Figure 14: Time-height cross sections of microwave radiometer temperature (upper panels), and radio acoustic sounding system virtual temperature and winds from the radar wind profiler (lower panels) at Wasco, OR, for January 4, 2017 (left) and January 19, 2017 (right). Red and blue arrows on the y-axes represent the sunrise and sunset times, respectively.**





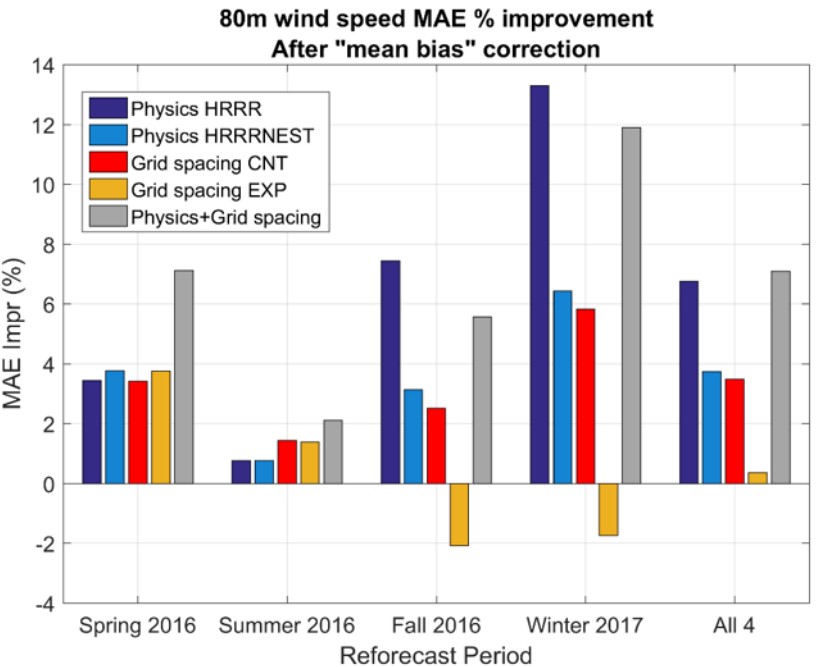

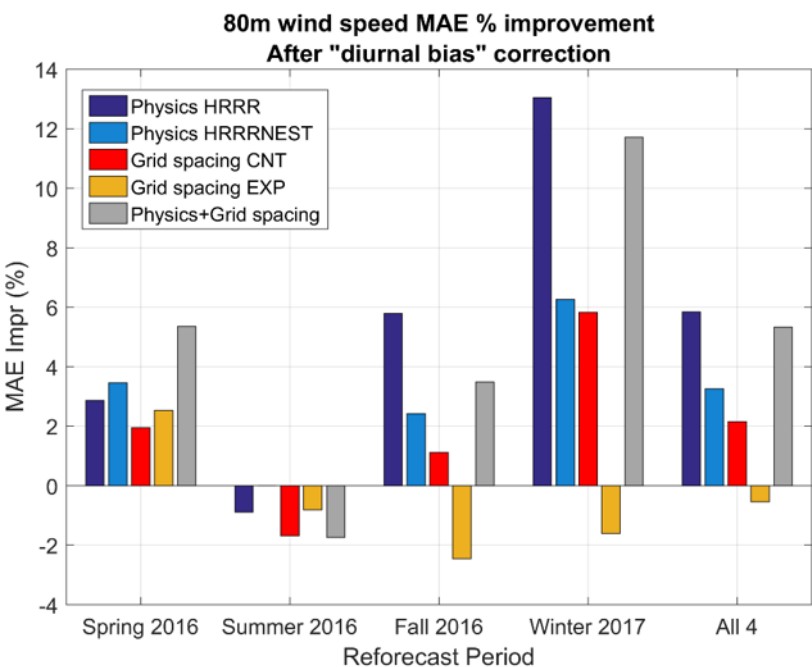

**Fig. 15. Percentage improvements on 80-m wind speed MAE (after bias correcting the model output) due to the experimental physics, finer horizontal grid spacing, and the combination of the two, for the four reforecast periods separately and averaged together. Upper panel: results using a "mean bias" correction; lower panel: results using a "diurnal bias" correction.**

