# Peer review of "Impact of model improvements on 80-m wind speeds during the second Wind Forecast Improvement Project (WFIP2)"

_Geoscientific Model Development, 2019_

## Referee Comment (RC1) · Anonymous Referee #1 · 28 Jun 2019

Review of the manuscript gmd-2019-80 Impact of model improvements on 80-m wind speeds during the second Wind Forecast Improvement Project (WFIP2) by Laura Bianco et al.

Summary

Within the context of the WFIP2 experiment, the authors evaluate the HRRR model on the performance for the 80 m wind speed. In addition they test whether a set of newly implemented physics schemes and/or increased spatial resolution improve the model performance. The evaluation covers multiple seasons, multiple starting times (Z00 and Z12) and is performed against a multiplicity of observational systems. In

general an increased resolution improves the model forecast, and the experimental physics suite is only beneficial in the HRRR, but not in the NEST version. Finally the authors unravel under which types of atmospheric phenomena the experiments result in a reduced or enlarged model bias. I find this study a very thorough evaluation that clearly illustrates the challenges the field faces when comparing and improving modelling systems, i.e. against different statistical metrics for different resolutions, under contrasting scale awareness of the model etc. However, I think the paper can be strengthened with limited amount of extra work in order to become more complete in terms of model variables and in terms of setting the future research agenda for model development.

Recommendation: major revisions

Major Remarks:

1. My first concern relates to the fact that this manuscript does not describe the physical package of EXP. The authors refer to earlier papers that document these modifications. While I understand the argument of doing so, as a reader I find it usually very unattractive to first read one or two other papers to understand the current one. SO I would encourage the authors to reserve some room to summarize the physical settings of EXP, so it becomes more clear to the reader what settings are underlying the bias reductions. I also think this helps the paper to generate more citations.

2. Although I understand that the focus of WFIP2 is on wind energy, it would be interesting for the readership to learn to what extent the model improvements also hold for wind speeds at other heights above the surface (60 m, 100m, 120 m – hub heights are rapidly increasing). One does not need to show all graphs for all heights, buts some guidance whether improved skill for the 80-m wind is also present at other levels is interesting for the readership of the paper.

3. In addition, it would be interesting to report whether improved statistics for wind also generate improved statistics for other variables as boundary-layer height, wind

direction, 10-m wind speed, 2m-temperature (let's say the routine synoptic variables). Again, no additional graphs are needed, but some guidance to know whether improved 80-m wind also improves or deteriorates the other variables is interesting to see the consistency of the improvements.

4. P6, ln 1: you suggest that the drag is too active in the revised physics. Is it possible to make this more concrete? E.g. one can discuss that this excess drag only occurs for grid cells where the modified drag scheme is active (since it switches on and off depending on the Froude number). Also if the PBL height in the model is too small, the drag has its divergence over a too shallow layer, making it too active in the atmosphere though the surface drag might be correct. In addition, it would be interesting to see whether one can distinguish whether the change in drag is due to local processes (surface drag) or modified synoptic settings induced indirectly by the drag.

5. I find the paper has a rather large amount of figures, while they are not always discussed in much depth. E.g. fig 14 can be removed, including the related text on P11, ln 1-18. In addition to that I would encourage the authors to extend the discussion about which atmospheric conditions are responsible for the model improvement. E.g. can the bias reduction be plotted against the geowind or vs atmospheric stability.

6. Although I appreciate the classification of the biases along different flow patterns, the exact definitions used to classify/categorize the flow patterns is missing in the paper. As such the reproducibility of the work is hampered.

7. Methodological concern: section starting at P11, ln 31: here the bias correction is applied and then it is concluded that the skills improves further. This is logical since you just removed the bias. A better way to do this is to split the data set in two parts and determine the bias correction on the first half and evaluate it independently on the second half of the data set. I could not understand from the paper whether this procedure was followed.

8. Finally: although I appreciate the efforts to report the model improvements and its

statistical evaluation, I think the paper can be strengthened by adding a section that summarizes the future research agenda concerning surface drag, the wind speed at hub heights. This is the journal of geoscientific model development, so in my opinion it should also prioritize the research efforts of the future.

Minor remarks:

P5, ln 7: when reading this I was wondering whether the statistics for other metrics behaved the same. This is dealt with later on in the paper, but perhaps it is good to announce already here that RMSE scores will be discussed later on. Just for the expectation management.

P5, ln 8: ... with SIGNIFICANTLY? smaller ...

P5, ln 11-15: this is a very long and unclear sentence

P7, ln 4-5: paragraph of 1 sentence, should be avoided

P7, ln 14: cite in chronological order.

P7, ln 18: .... always positive for wind speed.

P7, ln 24: model instead of models

P10, ln 12: reword "negative blue bar"

P10, ln 18-22: these sentences read like a figure caption, so is quite redundant

Figure 3: I would prefer to see this graph to be revised towards a column chart since the lines between the season do not say much. The statistics belong only to the season and are not connected.

[Figure]

---

## Short Comment (SC1) · 26 Jul 2019

Thank you for your efforts to provide the code and data underpinning this manuscript. there are just a couple of points on which you are not quite currently compliant with GMD policy and which need to be fixed in the revised manuscript.

GitHub URLs

GitHub is a great development platform, but it's not a suitable long term data archive. GitHub themselves tell you to use Zenodo for this purpose:

[Figure]

https://guides.github.com/activities/citable-code. Please use the instructions there to produce a Zenodo archive for the exact version of the code this manuscript describes and cite it. The Zenodo archive will even give you the BibTeX (or other reference manager) entry to paste in directly.

Data by URL

The WFIP2 data archive is impressively professional. However it is referenced by URL, which is unfortunate because URLs have a bad habit of going stale the next time an institution revamps its web presence. if at all possible, please provide a more persistent citation, such as a DOI, for this data.

For more precise details of what is required, please see https://www.geoscientific-model-development.net/about/code_and_data_policy.html

---

## Referee Comment (RC2) · Jeffrey Freedman (Referee) · 3 Aug 2019

This paper describes the results of model improvements to the High Resolution Rapid Refresh (HRRR) model developed using observations and improved parameterization schemes developed during the second Wind Forecast Improvement Project (WFIP2). Overall, the paper is very well organized with results presented in a clear and concise manner. The breakdown of model performance (e.g., improvement) by regime is especially noteworthy. This was an enjoyable paper to review and will be of great value to the observational and modeling communities.

General comments:

The manuscript refers to papers that are not yet available (e.g., Olsen et al. 2019a; McCaffrey et al. 2019). That made it problematic in reviewing the specifics regarding the differences between the HRRR CTL and EXP configurations (although the narrative does include parenthetical examples of the parameterizations/schemes that were modified). Although the other WFIP2 papers include a map of the instrument deployment/HRRR nests, if space were not an issue that would be helpful (readers, at times, are sometimes limited to printed versions). There are several examples of text in the narrative that are figure captions Some more speculation as to why (from a meteorological perspective) model performance categorized by regime differed by season (e.g. spring versus fall for gap flows and HRRR physics) would be of interest and value.

Specific comments:

Page 1 (Abstract), line 25: use of the word "versus"—perhaps should be consistent by just using "and." Page 1, line 34: "...also looking for the causes of model weaknesses" is a sentence fragment. Page 2, line 6: "hub-height" needs to be defined here (80 m given the other references). Page 4, line 11: more specificity on the spin-up problems with the HRRRNEST? Page 4, line 24: how "close" was the model layer to 80 m? Page 5, lines 6 - 7: "Initialization times....theZ00 and Z12 values." This is a figure caption. Page 6, lines 9 - 10: "Figure 3 displays...." Figure caption. Page 6, Figure 3: any difference (in relative magnitude) if %MAE was used? That is, larger errors during nocturnal period may have been due to higher wind speeds? Page 6, Figure 4: do higher elevations feature, on average, higher wind speeds? Perhaps a plot (or part of a plot) could show the diurnal average of the wind speeds for individual stations. Page 6, Figure 4: one station (ykm at 330 m) seems to have an unusually high bias—any explanation for this? Page 7, lines 7 - 8: "In this analysis...." This is interesting—a "decoupling" (assuming a well-mixed PBL over the region—not sure of this) of some sites at different times? Page 7, lines 21 - 26, sentence beginning "The upper panels display...." Figure caption. Page 7, lines 26 - 29: this is the only text describing Figure 5. Page 8, bottom lines, Figure 8: caption appears to be incomplete. It does not

mention this is for the combined impact. Page 10, line 9: "In truth, this figure does not tell the entire story." Literary flourish? Page 11, line 7: "...different atmospheric characteristics." In what way? On what scale. (At the bottom of this paragraph [lines 14 - 16] there is a mention of stability and wind profiles. Is this what is meant?)
* * *

---

## Referee Comment (RC3) · Jeffrey Freedman (Referee) · 3 Aug 2019

Note on Page 3, lines 14 - 17 contain a repetitive clause: ",...includes 3 449-MHz, 8 915-MHz radar wind profilers with radio acoustic sounding system temperature profiles, 19 sodars, 5 scanning lidars, 5 profiling lidars, 4 microwave radiometers, 10 microbarographs, a network of sonic anemometers, and many surface meteorological stations."
* * *

---

## Author Comment (AC1) · 9 Sep 2019

Review of the manuscript gmd-2019-80 Impact of model improvements on 80-m wind speeds during the second Wind Forecast Improvement Project (WFIP2) by Laura Bianco et al.

Summary
Within the context of the WFIP2 experiment, the authors evaluate the HRRR model on the performance for the 80 m wind speed. In addition they test whether a set of newly implemented physics schemes and/or increased spatial resolution improve the model performance. The evaluation covers multiple seasons, multiple starting times (Z00 and Z12) and is performed against a multiplicity of observational systems. In general an increased resolution improves the model forecast, and the experimental physics suite is only beneficial in the HRRR, but not in the NEST version. Finally the authors unravel under which types of atmospheric phenomena the experiments result in a reduced or enlarged model bias. I find this study a very thorough evaluation that clearly illustrates the challenges the field faces when comparing and improving modelling systems, i.e. against different statistical metrics for different resolutions, under contrasting scale awareness of the model etc. However, I think the paper can be strengthened with limited amount of extra work in order to become more complete in terms of model variables and in terms of setting the future research agenda for model development.
Recommendation: major revisions
We thank the Referee for the thoughtful comments. We hope we have addressed all of the Referee's concerns and we think that our manuscript did benefit from the constructive comments made by both Referees.

Major Remarks:

1. My first concern relates to the fact that this manuscript does not describe the physical package of EXP. The authors refer to earlier papers that document these modifications. While I understand the argument of doing so, as a reader I find it usually very unattractive to first read one or two other papers to understand the current one. So I would encourage the authors to reserve some room to summarize the physical settings of EXP, so it becomes more clear to the reader what settings are underlying the bias reductions. I also think this helps the paper to generate more citations.
We thank the Referee for the suggestion. We agree that having to read another paper to understand the current one is not appealing to any reader. In light of the comment from both Referees we decided to expand section 2.2 "NWP Models", including a list along with brief summaries of the complete set of model physical parameterizations and

relevant numerical methods targeted for development in WFIP2. We still refer to Olson et al. (2019a; 2019b), which in the meantime have been accepted for publication and are available (Olson et al. 2019a as early online releases), for details on the model configurations, but we hope this addition will give the reader all the needed tools for understanding the basic model settings used as part of this analysis.

2. Although I understand that the focus of WFIP2 is on wind energy, it would be interesting for the readership to learn to what extent the model improvements also hold for wind speeds at other heights above the surface (60 m, 100m, 120 m – hub heights are rapidly increasing). One does not need to show all graphs for all heights, buts some guidance whether improved skill for the 80-m wind is also present at other levels is interesting for the readership of the paper.

We certainly agree with the Referee on this matter. In fact, the dataset collected during WFIP2 is very rich and many other studies have been, or are being, performed to verify model improvements on other variables. While this paper was in revision other papers were accepted for publication, being submitted, or in preparation (for example a paper evaluating the models in the lowest 1 km of the atmosphere using data from scanning Doppler lidars at 3 sites is in preparation) on these other aspects and for this reason we don't think it is useful to repeat the analysis presented elsewhere in our paper. Nevertheless, in accordance with the Referee's comment, to give a wider view of the impact of the WFIP2 effort, we decided to expand Section 4 (adding subsection 4.6 "Impact of model improvements on other key meteorological variables") to summarize these other results.

3. In addition, it would be interesting to report whether improved statistics for wind also generate improved statistics for other variables as boundary-layer height, wind direction, 10-m wind speed, 2m-temperature (let's say the routine synoptic variables).Again, no additional graphs are needed, but some guidance to know whether improved80-m wind also improves or deteriorates the other variables is interesting to see the consistency of the improvements.

For boundary-layer height we are working on a separate manuscript that will focus on that aspect in particular, therefore we believe it is beyond the scope of the current work. For the other key meteorological variables mentioned by the Referee, as 10-m wind speed and 2-m temperature we did summarize the results found in Olson et al (2019) in the new subsection (4.6), as already mentioned in the answer to the comment above.

4. P6, ln 1: you suggest that the drag is too active in the revised physics. Is it possible to make this more concrete? E.g. one can discuss that this excess drag only occurs for grid cells where the modified drag scheme is active (since it switches on and off depending on the Froude number). Also if the PBL height in the model is too small, the drag has its divergence over a too shallow layer, making it too active in the atmosphere though the surface drag might be correct. In addition, it would be interesting to see whether one can distinguish whether the change in drag is due to local processes (surface drag) or modified synoptic settings induced indirectly by the drag.

There are two new sources of drag: the small-scale gravity wave drag (SSGWD) and the wind farm parameterization (WFP). The SSGWD is only active in the HRRR (for dx

> 1 km), so it does not contribute to the low near-surface wind speed biases in the HRRRNEST (dx=750 m). The WFP is active in both the HRRR and HRRRNEST. Combined, these two new sources of drag contribute to the low wind speed bias in the HRRR during the night (SSGWD is not active during the day), while the WFP can help contribute to the low wind speed bias for both the HRRR and HRRRNEST during the day or night.

The SSGWD was originally designed to only parameterize small-amplitude gravity waves, excited by rolling hills or similar types of terrain characterized by standard deviations of subgrid-scale terrain of < about 150 m. However, the original form of the SSGWD allowed the stress to be keep increasing as the standard deviation exceeded 150 m, which is common in the NorthWest US. This has been modified since the model code freeze.

Yes, low PBL height biases in the stable regime can cause excessive drag due to exaggerating the divergence of the momentum stress. To limit this, within the SSGWD only, we assume that momentum stresses decrease to zero no lower than 300 m, so the SSGWD drag is always spread over a layer at least 300 m deep. We think this is reasonable, since small-scale gravity waves may propagate into the stable layer above the model-defined PBL height and may not break until they reach the more neutral residual layer immediately above the surface stable layer. This does result in some unwanted limits in the model code, but it helps to remove excessive drag that may be caused by poorly estimated PBL heights in the stable layer.

It may be interesting to better distinguish the drag due to local (surface frictional and/or form drag) *vs* the regional or synoptically modified flows indirectly caused by the drag, but since these new forms of drag typically only directly impact the lowest 300 m and typically only combine to provide a deceleration of the low-level winds between 0.1-0.5 m s-1, we suspect that the effect on the synoptic scale is very small for forecasts less than 24 hrs in length. Also, the investigation may be contaminated by the lateral boundary conditions needed in limited area modeling. Therefore, we think this extra exercise to distinguish the drag effects from local *vs* synoptic are best suited for medium range forecasts (5-10 days) within a global modeling framework.

5. I find the paper has a rather large amount of figures, while they are not always discussed in much depth. E.g. fig 14 can be removed, including the related text on P11, ln 1-18.

According to the Referee's suggestion Fig. 14 was removed from the revised version of the manuscript. Some discussion on the behavior of the models due to the different characteristics of the cold pool events highlighted in Fig. 13 remains in the text nonetheless.

In addition to that I would encourage the authors to extend the discussion about which atmospheric conditions are responsible for the model improvement. E.g. can the bias reduction be plotted against the geowind or vs atmospheric stability?

Unfortunately, we do not have observed geostrophic wind or atmospheric stability at all the sites used in this study. Some of the sites have co-located radar wind profilers (not all, though) but the maximum height reached by this instrument is well below the geostrophic wind level. Atmospheric stability could be derived at some of the sites,

where microwave radiometers are available, but these are only 3 out of the 22 used in our study, making that variable non-representative of the entire area of interest. In any case, since Fig. 1, 2, 5, 6, 7, and 8 present the statistics as a function of the time of the day, we believe that some insight about what atmospheric conditions are responsible for the model improvements could be derived by these. Therefore, according to the Referee's suggestion we pointed to the dependence on atmospheric stability in the revised version of the manuscript, specifically, where we discussed the above-mentioned figures.

6. Although I appreciate the classification of the biases along different flow patterns, the exact definitions used to classify/categorize the flow patterns is missing in the paper. As such the reproducibility of the work is hampered.

We understand the Referee's comment on the lack of the exact definitions used to differentiate between the different flow patterns. At the beginning of the campaign several meetings were organized between the meteorologists that volunteered to participate in the weather discussions to the purpose of the creation of the Event Log. The classifications were based on the available observations, operational analysis products, HRRR forecasts, satellite images, and local radiosondes. Due to the fact that not all of these were available at all times, it was not possible to base classifications on specific thresholds and definitions. It was certainly a process that involved a certain level of subjectivity, as we already pointed out in the manuscript. Nevertheless, the process involved weekly meetings during the field study with meteorologists on the project team, many with operational forecasting experience in this geographic area, during which a consensus was reached by the team, making us confident that other meteorologists would agree with the classifications we used. Also, the Event Log is accessible to the public (available on the DAP, https://a2e.energy.gov/projects/wfip2), so the reproducibility of the work is not hampered in this sense. Some additional text was added to the revised version of the manuscript about this.

7. Methodological concern: section starting at P11, ln 31: here the bias correction is applied and then it is concluded that the skills improves further. This is logical since you just removed the bias. A better way to do this is to split the data set in two parts and determine the bias correction on the first half and evaluate it independently on the second half of the data set. I could not understand from the paper whether this procedure was followed.8.

We thank the Referee for this suggestion. According to his comment we have modified the procedure used to apply the bias correction. We now split the dataset into two parts, determine the bias correction to apply from the first part and evaluate it independently on the second half of the data set.

Finally: although I appreciate the efforts to report the model improvements and its statistical evaluation, I think the paper can be strengthened by adding a section that summarizes the future research agenda concerning surface drag, the wind speed at hub heights. This is the journal of geoscientific model development, so in my opinion it should also prioritize the research efforts of the future.

Since the model code freeze, we have prioritized three research tasks related to better simulating the low-level wind speeds: (1) the inclusion of momentum transport in the new mass-flux component of the MYNN-EDMF (already completed), (2) modifying the SSGWD to only parameterize small-amplitude gravity waves associated with subgrid-scale terrain undulations < 100 m (also completed), and (3) investigating the addition of a vertically distributed form drag as opposed to represent form drag only through the surface roughness length, which is probably only valid for dx < 1 km, where the terrain is better resolved. The impact of (1) tends to increase the near-surface wind speed in the convective boundary layer, which helps to correct the low wind speed bias we measured in WFIP2. Tasks (2) and (3) are simply meant to revise the original representation of drag in the HRRR in order to make the parameterizations more physically meaningful. All of these model components need to be investigated at a variety of model resolutions spanning dx = 1 to 10 km to ensure the model parameterizations successfully adapt in behavior to only represent the physical processes that are truly not well-resolved within the model.

Minor remarks:

P5, ln 7: when reading this I was wondering whether the statistics for other metrics behaved the same. This is dealt with later on in the paper, but perhaps it is good to announce already here that RMSE scores will be discussed later on. Just for the expectation management.
We don't look at RMSE in this study, but mostly at MAE and biases and this is pointed out in the text.

P5, ln 8: ... with SIGNIFICANTLY? smaller ...
Done.

P5, ln 11-15: this is a very long and unclear sentence.
We reworded the sentence as: "*Figure 1 can be used to examine the dependence of MAE on initialization time and forecast horizon. In particular, the Z00 MAEs are smaller than the Z12 MAE values for times soon after the Z00 initialization (for the first part of the day O lines are below X lines). In contrast the Z12 MAEs tend to be smaller than Z00 values for times soon after the Z12 initialization (for the second part of the day X lines are below O lines, except for HRRRNEST EXP), meaning that the MAE increases with the forecast horizon.*"

P7, ln 4-5: paragraph of 1 sentence, should be avoided.
Done.

P7, ln 14: cite in chronological order.
Done.

P7, ln 18: .... always positive for wind speed.
Done.

P7, ln 24: model instead of models
Done.

P10, ln 12: reword "negative blue bar"
Done. The sentence has been reworded from: "*the negative blue bar in spring and summer, visible in Fig. 9…*" to: "*blue bar in spring and summer extending toward negative values, visible in Fig. 9…*"

P10, ln 18-22: these sentences read like a figure caption, so is quite redundant
According to the Referee's suggestion we removed the sentence "*HRRR CNT is shown in red, HRRR EXP is in blue, and observations are in black. In the lower panel, gap flow days are highlighted with the red shaded areas.*"

Figure 3: I would prefer to see this graph to be revised towards a column chart since the lines between the seasons do not say much. The statistics belong only to the season and are not connected.
According to the Referee's suggestion Fig.3 has been modified into a bar chart.

---

## Author Comment (AC2) · 9 Sep 2019

Jeffrey Freedman (Referee) jfreedman@albany.edu

This paper describes the results of model improvements to the High Resolution Rapid Refresh (HRRR) model developed using observations and improved parameterization schemes developed during the second Wind Forecast Improvement Project (WFIP2).Overall, the paper is very well organized with results presented in a clear and concise manner. The breakdown of model performance (e.g., improvement) by regime is especially noteworthy. This was an enjoyable paper to review and will be of great value to the observational and modeling communities.

We thank Dr. Freedman for offering his opinion on our manuscript. We appreciate his thoughtful comments. We hope we have addressed all of the Referee's concerns and we think that our manuscript did benefit from the constructive comments made by both Referees.

General comments:

The manuscript refers to papers that are not yet available (e.g., Olsen et al. 2019a; McCaffrey et al. 2019). That made it problematic in reviewing the specifics regarding the differences between the HRRR CTL and EXP configurations (although the narrative does include parenthetical examples of the parameterizations/schemes that were modified).

Based on  the comments from both Referees we decided to expand section 2.2 "NWP Models" to include a list with brief summaries of the complete set of model physical parameterizations and relevant numerical methods targeted for development in WFIP2. We still refer to Olson et al. (2019a; 2019b), which in the meantime have been accepted for publication and are available (Olson et al. 2019a as early online releases), for accurate details on the improved model configurations, but we hope this addition will give the reader all the needed tools for understanding the basic settings of the models' runs.

Although the other WFIP2 papers include a map of the instrument deployment/HRRR nests, if space were not an issue that would be helpful (readers, at times, are sometimes limited to printed versions).

We thank the Referee for the suggestion. According to the Referee's comment a topographic map, with the location of the sites, has been inserted as a new panel in Fig. 4. We hope that this and other additions we incorporated into the manuscript (see answer to the comment above) will make the paper more self-consistent.

There are several examples of text in the narrative that are figure captions.

We modified the text in the revised version of the manuscript when this was pointed out by the Referee.

Some more speculation as to why (from a meteorological perspective) model performance categorized by regime differed by season (e.g. spring versus fall for gap flows and HRRR physics) would be of interest and value.
Gap flow events are of different nature over different seasons. From our analysis it seems that in summer, thermally forced gap flow are problematic and difficult to forecast, but in winter, synoptically forced gap flows show an improvement in the model forecast. Some text about this has been added in the revised version of the manuscript.

Specific comments:

Page 1 (Abstract), line 25: use of the word "versus" perhaps should be consistent by just using "and." Page 1, line 34: ". . . also looking for the causes of model weaknesses" is a sentence fragment.
The word "*versus*" in the Abstract was changed to "*and*". Also, the sentence "*...also looking for the causes of model weaknesses*" was changed to "*Causes of model weaknesses are identified*"

Page 2, line 6: "hub-height" needs to be defined here (80 m given the other references).
Done.

Page 4, line 11: more specificity on the spin-up problems with the HRRRNEST?
The 3-km HRRR is directly initialized off of the 13-km RAP grid, so there is a spin-up period associated with the model atmosphere adjusting to the higher resolution terrain, which typically has much higher mountain peaks and lower valleys in the HRRR relative to the RAP. This spin-up problem would be even more exaggerated if the HRRRNEST was directly initialized from the RAP model atmosphere, so to minimize this problem, we chose to allow the HRRR model atmosphere to spin-up for 3 hrs before we initialized the HRRRNEST from the HRRR 3-hr forecast. New text has been added to the revised manuscript to clarify this issue.

Page 4, line 24: how "close" was the model layer to 80 m?
The text has been modified to include this info as: "*For our analysis, in order to compare to the observations, the 80-m wind field is obtained from model output horizontally bi-linearly interpolating to the 22 site locations using the 4 closest grid points, and linearly vertically interpolating the two closest heights (approximately 36 and 83 m).*"

Page5, lines 6 - 7: "Initialization times . . . .theZ00 and Z12 values." This is a figure caption.
The text has been changed in the revised version of the manuscript to "*Initialization times are represented with the O's (Z00 runs) and with the X's (Z12 runs), while the averages between these values are in solid, bold lines.*"

Page 6, lines 9 - 10: "Figure 3 displays . . . ." Figure caption.

The text has been changed in the revised version of the manuscript to hopefully read less as a figure caption ("*MAEs of the 80-m wind speed, presented in the left panel of Fig. 3, show that the HRRR EXP (in blue) does better than the HRRR CNT (in red) in fall and in winter, but not in spring nor summer. MAEs of the HRRRNEST CNT (in yellow) are better than those of the HRRR CNT (in red), and the HRRRNEST EXP (in black) is now almost always better than the other models. Biases, presented on the right panel of Fig. 3, show values in the HRRR EXP (in blue) becoming way too negative (caused by the additional orographic drag employed in the HRRR EXP) compared to the HRRR CNT (in red) in the spring, summer and fall.*").

Page 6, Figure 3: any difference (in relative magnitude) if %MAE was used? That is, larger errors during nocturnal period may have been due to higher wind speeds?
We thank the Referee for making this good point. We think including the observed averaged diurnal 80-m wind speed cycle is important, but since adding a new figure was not an option due to the already large number of figures in the manuscript, we decided to include in the revised version of the manuscript an insert to panel *a* of Fig. 1, with the diurnal cycle of the averaged observed 80-m wind speeds for the four reforecast periods for reference. This new insertion shows how wind speeds are larger at nighttime, particularly in summer and to a lesser extent in spring, but less so in fall and winter. We also included some text regarding this in the revised version of the manuscript when discussing the magnitudes of the errors for the different periods (Sec 3.1: "*For reference, the insert of panel a of Fig. 1 presents the diurnal cycle of the averaged observed 80-m wind speeds for the four reforecast periods, showing that 80-m wind speeds are higher at nighttime, particularly in summer and to a lesser extent in spring (contributing to MAE to be larger at nightime compared to daytime), but less so in fall and winter.*").
To address the Referee's comments (both the current and the next comment) we made a plot (shown below but not included in the manuscript), for the four reforecast periods separately, with the averaged observed 80-m wind speeds at all sites (black line) and over an average at three elevation ranges:
- 0-300 m: AON3, AON7, BOR, RFS, ARL (in blue),
- 300-700 m: AON2, AON4, AON5, GDL, WCO, WWL, YKM, VCR (in green),
- > 700 m: AON1, AON6, AON8, AON9, CDN, DCR, PVE, RTK, GDR (in red).

[Figure]

From this figure we see similar diurnal patterns for wind speed for all three elevation ranges, but more interesting is to notice that, while the sites with lower elevation (blue and green lines) experience stronger 80-m wind speeds compared to those at higher elevation (red line) in summer, for fall and winter the opposite is true. This might be due to gap flow events happening more often in summer, and cold pool events, with lower wind speeds closer to the surface and higher wind speeds above, happening more often in fall and winter. Spring does not show much difference in the diurnal behavior of 80-m wind speeds for sites at different elevations.

Page 6, Figure 4: do higher elevations feature, on average, higher wind speeds? Perhaps a plot (or part of a plot) could show the diurnal average of the wind speeds for individual stations.

To address this Referee's question relative to Fig. 4, we added (as a dotted black line) the averaged 80-m wind speed at each site for the 4 reforecast periods (panels *a* to *d* of Fig. 4). To include these extra lines we incorporated right axes to each panel. These added lines can be used to answer the Referee's comment. Specifically there is some dependence of 80-m wind speed with site elevation in fall and winter, most likely caused by cold pool events with lower wind speeds confined to lower elevations. On the other side, and also according to the figure in the answer to the comment above, we see that sites at higher elevations do not show higher 80-m wind speeds compared to that of sites at lower elevations neither in summer nor in spring.

Page 6, Figure 4: one station (ykm at 330 m) seems to have an unusually high bias any explanation for this?

The Yakima site is the one to the farthest North in the study area, as visible form the new *e* panel of Fig. 4 (included in the revised version of the manuscript). Forecasts at

this site are particularly difficult due to the presence of the developed area, (on the North-East of the site), crops on its South-East, and a very steep ridge on its south. While the elevation of the site is at ~330m, the top of the ridge is at double this elevation. These features are visible in the map below.
This is a challenging location for models to get the details correct. Also, the ridge separates Yakima from the main study area, which could lead to different results.

[Figure]

Page 7, lines 7 - 8: "In this analysis . . . ." This is interesting a "decoupling" (assuming a well-mixed PBL over the region not sure of this) of some sites at different times?
We think the text the Referee is referring to our statement that "*Terrain complexity is not as powerful of a predictor of model bias as site elevation. A similar analysis to that presented in Fig. 4 was performed but sorting the sites by the complexity of the surrounding terrain (see Table 1). In this analysis (not shown) the trend of 80-m wind speed MAE and bias was not clearly defined.*" The point we are attempting to make is that using the complexity of the terrain surrounding the sites to sort the elements on the x axes we do not see a well-defined trend in neither MAE nor bias. But we do not know what kind of decoupling we can make responsible for this; therefore, we did not change the text in the revised version of the manuscript.

Page 7, lines 21 - 26, sentence beginning "The upper panels display . . . ." Figure caption.

Text in the revised version of the manuscript has been modified to read less as a figure caption.

Page 7, lines 26 - 29: this is the only text describing Figure5.

Figure 5 is described in the entire section 4.1.

Page 8, bottom lines, Figure 8: caption appears to be incomplete. It does not mention this is for the combined impact.

Thanks to the Referee for catching this oversight. The label of Fig 8 has been modified to: "*As in Fig. 6 but for HRRRNEST EXP (in black) vs HRRR CNT (in red) runs, showing the combined impact on 80-m wind speed MAE of the experimental physics and finer model horizontal grid spacing.*"

Page 10, line 9: "In truth, this figure does not tell the entire story." Literary flourish?

We are trying to keep the reader's interest up at this point!….

Page 11, line 7: ". . . different atmospheric characteristics." In what way? On what scale. (At the bottom of this paragraph [lines 14 - 16] there is a mention of stability and wind profiles. Is this what is meant?)

Yes, we did use the time-height cross sections of microwave radiometer temperature, winds from the radar wind profiler, and radio acoustic sounding system virtual temperature to find that the cold pool at the beginning of January is brought in by sustained easterly winds and has weaker stable stratification compared to the cold pool event in the second half of January, which is characterized by very low wind speeds close to the surface and more strongly stable stratification. According to the suggestion of Referee #1 Fig. 14 (presenting these time-height cross sections) was removed from the revised version of the manuscript, some discussion on the behavior of the models due to the different atmospheric characteristics of the cold pool events highlighted in Fig. 13 is nonetheless still discussed in the text.

Note on Page 3, lines 14 - 17 contain a repetitive clause: ",...includes 3 449-MHz, 8915-MHz radar wind profilers with radio acoustic sounding system temperature profiles,19 sodars, 5 scanning lidars, 5 profiling lidars, 4 microwave radiometers, 10 microbaro-graphs, a network of sonic anemometers, and many surface meteorological stations."

We thank the Referee for catching the repetition, which has now been removed.

---

## Author Comment (AC3) · 9 Sep 2019

David Ham david.ham@imperial.ac.uk

Thank you for your efforts to provide the code and data underpinning this manuscript. There are just a couple of points on which you are not quite currently compliant with GMD policy and which need to be fixed in the revised manuscript.

GitHub URLs

GitHub is a great development platform, but it's not a suitable long term data archive. GitHub themselves tell you to use Zenodo for this purpose: https://guides.github.com/activities/citable-code. Please use the instructions there to produce a Zenodo archive for the exact version of the code this manuscript describes and cite it. The Zenodo archive will even give you the BibTeX (or other reference manager) entry to paste in directly.

According to the Editor suggestion we did produce a Zenodo archive for the model code (https://zenodo.org/record/3369984#.XVb6KpJKjUI, doi:10.5281/zenodo.3369984). The text in the "Data and code availability" section has been changed accordingly.

Data by URL

The WFIP2 data archive is impressively professional. However it is referenced by URL, which is unfortunate because URLs have a bad habit of going stale the next time an institution revamps its web presence. If at all possible, please provide a more persistent citation, such as a DOI, for this data. For more precise details of what is required, please see https://www.geoscientific-model-development.net/about/code_and_data_policy.htm

Unfortunately at this time a DOI number is not available for the DAP, but the archive is maintained through the Atmosphere to Electrons (A2e) research and development initiative, supported by the U.S. Department of Energy, Office of Energy Efficiency and Renewable Energy's Wind Energy Technologies Office to the aim of collect, store, and preserve A2e data. So, even if we are not directly in charge of the repository, we do not foresee a revamp of its web presence. All data from A2e projects are stored and disseminated via the A2e Data Archive and Portal at a2e.energy.gov and also other WFIP2 publications referred to the DAP by URL.

---

## Author Comment (AC4) · 9 Sep 2019

[revised manuscript text omitted]

Sum 16: 96
Fall 16: 91
Win 17: 33 | Vaisala |
| sodar | AON2 | 45.554 | 120.156 | 356 | 13 | Spr 16: 98
Sum 16: 98
Fall 16: 93
Win 17: 94 | Vaisala |
| sodar | AON3 | 45.938 | 119.406 | 116 | 12 | Spr 16: 97
Sum 16: 98
Fall 16: 92
Win 17: 84 | Vaisala |
| sodar | AON4 | 45.637 | 120.680 | 432 | 34 | Spr 16: 98
Sum 16: 97
Fall 16: 92
Win 17: 72 | Vaisala |
| sodar | AON5 | 45.575 | 120.747 | 456 | 13 | Spr 16: 99
Sum 16: 99
Fall 16: 93
Win 17: 95 | Vaisala |
| sodar | AON6 | 45.516 | 120.781 | 731 | 81 | Spr 16: 97
Sum 16: 84
Fall 16: 82
Win 17: 89 | Vaisala |
| sodar | AON7 | 45.631 | 121.069 | 166 | 55 | Spr 16: 97
Sum 16: 16
Fall 16: 0
Win 17: 86 | Vaisala |
| sodar | AON8 | 45.602 | 121.589 | 703 | 98 | Spr 16: 34
Sum 16: 0
Fall 16: 0
Win 17: 0 | Vaisala |
| sodar | AON9 | 45.374 | 121.330 | 836 | 57 | Spr 16: 0
Sum 16: 0
Fall 16: 0
Win 17: 51 | Vaisala |
| sodar | BOR | 45.816 | 119.812 | 112 | 6 | Spr 16: 95 | NOAA/ARL |

| | | | | | | Sum 16: 96
Fall 16: 74
Win 17: 83 | |
|---|---|---|---|---|---|---|---|
| sodar | CDN | 45.245 | 120.169 | 891 | 25 | Spr 16: 8
Sum 16: 37
Fall 16: 84
Win 17: 97 | DOE/NREL |
| sodar | DCR | 45.165 | 120.656 | 795 | 26 | Spr 16: 96
Sum 16: 98
Fall 16: 97
Win 17: 92 | DOE/NREL |
| sodar | GDL | 45.805 | 120.849 | 501 | 16 | Spr 16: 95
Sum 16: 98
Fall 16: 90
Win 17: 87 | DOE/ANL |
| sodar | PVE | 44.285 | 120.901 | 991 | 42 | Spr 16: 96
Sum 16: 96
Fall 16: 92
Win 17: 57 | NOAA/ARL |
| sodar | RFS | 45.691 | 120.746 | 62 | 80 | Spr 16: 48
Sum 16: 4
Fall 16: 11
Win 17: 23 | UND |
| sodar | RTK | 45.364 | 120.747 | 708 | 19 | Spr 16: 94
Sum 16: 98
Fall 16: 89
Win 17: 41 | DOE/PNNL |
| sodar | WCO | 45.590 | 120.672 | 462 | 25 | Spr 16: 81
Sum 16: 88
Fall 16: 69
Win 17: 71 | NOAA/ARL |
| sodar | WWL | 46.095 | 118.261 | 382 | 34 | Spr 16: 91
Sum 16: 85
Fall 16: 83
Win 17: 97 | DOE/ANL |
| sodar | YKM | 46.572 | 120.551 | 330 | 19 | Spr 16: 96
Sum 16: 73
Fall 16: 25
Win 17: 85 | DOE/ANL |
| scanning | ARL | 45.720 | 120.187 | 266 | 56 | Spr 16: 100 | NOAA/ESRL |

| | | | | | | | |
|---|---|---|---|---|---|---|---|
| lidar | | | | | | Sum 16: 100
Fall 16: 28
Win 17: 95 | |
| profiling lidar | GDR | 45.516 | 120.780 | 725 | 81 | Spr 16: 90
Sum 16: 90
Fall 16: 71
Win 17: 0 | CU |
| profiling lidar | VCR | 45.954 | 118.688 | 542 | 69 | Spr 16: 93
Sum 16: 97
Fall 16: 78

[revised manuscript text omitted]

---

## Author Comment (AC6) · 9 Sep 2019

The comment was uploaded in the form of a supplement:
https://www.geosci-model-dev-discuss.net/gmd-2019-80/gmd-2019-80-AC6-supplement.pdf

---

## Author Comment (AC7) · 9 Sep 2019

The comment was uploaded in the form of a supplement: https://www.geosci-model-dev-discuss.net/gmd-2019-80/gmd-2019-80-AC7-supplement.pdf

---

## Author Comment (AC8) · 9 Sep 2019

The comment was uploaded in the form of a supplement:
https://www.geosci-model-dev-discuss.net/gmd-2019-80/gmd-2019-80-AC8-supplement.pdf

---

## Author Response (AR2)

Comments to the Author:

Dear authors,

your paper is almost ready for publication. However, I would like you to make a few changes before I accept the manuscript.

We thank the Editor for carefully checking the manuscript for places where the text was not clear yet. We hope to have included all the modifications, as requested.

We also noticed that we had missed to list Benjamin et al. (2016) among the list of References, so we included it in the revised version of the manuscript.

Finally, we noticed that the site "GDL" in panel *e* of Fig. 4 was misspelled to "DGL", so this panel has been updated to a new one, where the misspell was fixed, in the revised version of the manuscript.

1) Sometimes you use the word "verification" or indeed "model verification". This is a strong word and often misused as it implies that somebody knows the truth. In relation to models it is patently inappropriate since a model is only a more or less useful tool to SIMULATE nature. Thus, please check your use of "verification" and change to other expressions where you feel that the truth is not actually known. "Model verification" must be changed to something more appropriate.

We modified the text according to the Editor's suggestion, removing the word "verification" completely in the revised version of the manuscript.

2) P 6, L 13: instead of "improves a .. bias" please write "reduces ... bias" and check whether your "low" wind speed bias isn't rather a negative wind speed bias.

We modified the text according to the Editor's suggestion in the revised version of the manuscript.

3) P 9, L 5: please write "smaller overall bias" instead of "better overall bias".

Done.

4) P 12, L 7-8: "positive improvements" -> "improvements"

Done.

5) P 12, L 15: "Negative impacts to the improvement", please reword!

We changed "Negative impacts to the improvement" to "Degradations" in the revised version of the manuscript.

6), P 13, L 7: Delete "In truth" and simply start the sentence with "This figure" or replace "In truth" with "However".

We changed "In truth" to "However," in the revised version of the manuscript.

7) P 15, L 20: "improvements ... were much smaller...", much smaller than what? I assume you mean

much smaller than at 80 m. In this case, can you explain or do you have a conjecture why the improvements are smaller at other altitudes than at 80 m? What is peculiar with just 80 m?

We changed the text to hopefully make this sentence clearer in the revised version of the manuscript.

8) P 15, L 26: spell out CONUS.

Done.

[revised manuscript text omitted]

Sum 16: 96
Fall 16: 91
Win 17: 33 | Vaisala |
| sodar | AON2 | 45.554 | 120.156 | 356 | 13 | Spr 16: 98
Sum 16: 98
Fall 16: 93
Win 17: 94 | Vaisala |
| sodar | AON3 | 45.938 | 119.406 | 116 | 12 | Spr 16: 97
Sum 16: 98
Fall 16: 92
Win 17: 84 | Vaisala |
| sodar | AON4 | 45.637 | 120.680 | 432 | 34 | Spr 16: 98
Sum 16: 97
Fall 16: 92
Win 17: 72 | Vaisala |
| sodar | AON5 | 45.575 | 120.747 | 456 | 13 | Spr 16: 99
Sum 16: 99
Fall 16: 93
Win 17: 95 | Vaisala |
| sodar | AON6 | 45.516 | 120.781 | 731 | 81 | Spr 16: 97
Sum 16: 84
Fall 16: 82
Win 17: 89 | Vaisala |
| sodar | AON7 | 45.631 | 121.069 | 166 | 55 | Spr 16: 97
Sum 16: 16
Fall 16: 0
Win 17: 86 | Vaisala |
| sodar | AON8 | 45.602 | 121.589 | 703 | 98 | Spr 16: 34
Sum 16: 0
Fall 16: 0
Win 17: 0 | Vaisala |
| sodar | AON9 | 45.374 | 121.330 | 836 | 57 | Spr 16: 0
Sum 16: 0
Fall 16: 0
Win 17: 51 | Vaisala |
| sodar | BOR | 45.816 | 119.812 | 112 | 6 | Spr 16: 95 | NOAA/ARL |

| | | | | | | Sum 16: 96
Fall 16: 74
Win 17: 83 | |
|---|---|---|---|---|---|---|---|
| sodar | CDN | 45.245 | 120.169 | 891 | 25 | Spr 16: 8
Sum 16: 37
Fall 16: 84
Win 17: 97 | DOE/NREL |
| sodar | DCR | 45.165 | 120.656 | 795 | 26 | Spr 16: 96
Sum 16: 98
Fall 16: 97
Win 17: 92 | DOE/NREL |
| sodar | GDL | 45.805 | 120.849 | 501 | 16 | Spr 16: 95
Sum 16: 98
Fall 16: 90
Win 17: 87 | DOE/ANL |
| sodar | PVE | 44.285 | 120.901 | 991 | 42 | Spr 16: 96
Sum 16: 96
Fall 16: 92
Win 17: 57 | NOAA/ARL |
| sodar | RFS | 45.691 | 120.746 | 62 | 80 | Spr 16: 48
Sum 16: 4
Fall 16: 11
Win 17: 23 | UND |
| sodar | RTK | 45.364 | 120.747 | 708 | 19 | Spr 16: 94
Sum 16: 98
Fall 16: 89
Win 17: 41 | DOE/PNNL |
| sodar | WCO | 45.590 | 120.672 | 462 | 25 | Spr 16: 81
Sum 16: 88
Fall 16: 69
Win 17: 71 | NOAA/ARL |
| sodar | WWL | 46.095 | 118.261 | 382 | 34 | Spr 16: 91
Sum 16: 85
Fall 16: 83
Win 17: 97 | DOE/ANL |
| sodar | YKM | 46.572 | 120.551 | 330 | 19 | Spr 16: 96
Sum 16: 73
Fall 16: 25
Win 17: 85 | DOE/ANL |
| scanning | ARL | 45.720 | 120.187 | 266 | 56 | Spr 16: 100 | NOAA/ESRL |

| | | | | | | Sum 16: 100 | |
|---|---|---|---|---|---|---|---|
| lidar | | | | | | Fall 16: 28 | |
| | | | | | | Win 17: 95 | |
| profiling lidar | GDR | 45.516 | 120.780 | 725 | 81 | Spr 16: 90 Sum 16: 90 Fall 16: 71 Win 17: 0 | CU |
| profiling lidar | VCR | 45.954 | 118.688 | 542 | 69 | Spr 16: 93 Sum 16: 97 Fall 16: 78 
[revised manuscript text omitted]